# NEURAL EMBEDDINGS FOR TEXT

## ABSTRACT

We propose a new kind of embedding for natural language text that deeply represents semantic meaning. Standard text embeddings use the outputs from hidden layers of a pretrained language model. In our method, we let a language model learn from the text and then literally pick its brain, taking the actual weights of the model's neurons to generate a vector. We call this representation of the text a neural embedding. We confirm the ability of this representation to reflect semantics of the text by an analysis of its behavior on several datasets, and by a comparison of neural embedding with state of the art sentence embeddings.

## 1 INTRODUCTION

Capturing the semantic meaning of text as a vector is a fundamental challenge for natural language processing (NLP) and an area of active research (Giorgi et al., 2021; Zhang et al., 2020; Gao et al., 2021; Huang et al., 2021; Yan et al., 2021; Zhang et al., 2021; Muennighoff, 2022; Alexander Liu, 2022; Chuang et al., 2022). Recent work has focused on fine-tuning pretrained language models with contrastive learning, either supervised (e.g. Reimers & Gurevych (2019); Zhang et al. (2021); Yan et al. (2021)) or unsupervised (e.g. Giorgi et al. (2021); Gao et al. (2021)). The embedding is generated by pooling the outputs of certain layers of the model as it processes a text.

Motivated by the need for deeper semantic representations of text, we propose a different kind of embedding. We allow a language model to fine-tune on a text input, and then measure the resulting changes to the model's own neuronal weights as a *neural embedding*. We demonstrate that neural embeddings do indeed represent the semantic differences between samples of text. We evaluate neural embeddings on several datasets and compare them with several state of the art sentence embeddings. We observe that neural embeddings correlate better specifically with semantics, while being comparable in other evaluations. We find that neural embeddings behave differently from the known embeddings we considered. Our contribution:

1. We propose a new kind of text representation: *neural embeddings*[1] (Section 2).
2. We evaluate embeddings by using several datasets and several criteria (Section 3). We show that by these criteria the neural embeddings are (1) better correlated with semantic similarity and consistency, and (2) strongly differ by the errors they do and by how they represent the qualities of the text.

## 2 NEURAL EMBEDDING METHOD

The technique for generating neural embeddings is using *micro-tuning*, first introduced for the BLANC-tune method of document summary quality evaluation[2] Vasilyev et al. (2020). It is a tuning on one sample only, and the tuned model is used for the sample only. Tuning a pretrained model on a specific narrow domain is a common practice to improve performance. Micro-tuning takes this to extreme, narrowing down the 'domain' to a 'dataset' consisting of just one sample.

For each text sample, we start with the original language model and fine-tune only a few selected layers $L_0, L_1, ..., L_m$ while keeping all other layers frozen. Once the fine-tuning on the text sample is complete, we measure the difference between the new weights $W'_j$ and the original weights $W_j$

---

[1]github url will be provided here. The code is in supplementary material.
[2]https://github.com/PrimerAI/blanc

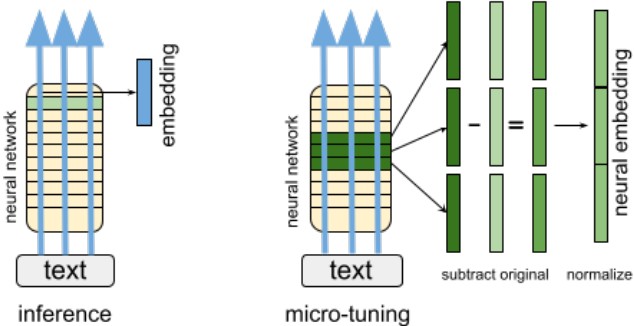

Figure 1: Illustration comparing the usual output embeddings (left) and neural embeddings (right). The output embeddings are taken from aggregated outputs of certain layers at inference, as shown on the left. Neural embeddings are taken from weights of certain layers at micro-tuning.

of each layer $L_j$ and normalize the resulting vector. We obtain the neural embedding of the text by concatenating the normalized vectors:

$$E = \frac{E_c}{|E_c|} , \quad E_c = \|_{j=0}^m (W_j' - W_j)/|W_j' - W_j| \tag{1}$$

Here the symbol $\|$ means concatenation, e.g. $\|_{j=1}^m a_j$ is a concatenation of $a_0, a_1, \dots, a_m$. Schematically, our illustration is in Figure 1. For clarity, the algorithm is shown in Appendix A, Figure 8.

For example, if we select three layers from the standard BERT base model, and if each selected layer has 768 weights, then the resulting embedding will have size $768 * 3 = 2304$. Before entering Equation 1, the weights of each layer are flattened from their possibly multidimensional tensor form.

Through this paper we use the pretrained transformer Devlin et al. (2019) model *bert-base-uncased* from the transformers library Wolf et al. (2020). We found that layers either from the top of the model, or from the last transformer block of the model perform best. In the next section we use the following selection (see Appendix A), by the notations of the huggingface transformers library [3]:

1. $L_0 = cls.predictions.transform.LayerNorm.weight$
2. $L_1 = cls.predictions.transform.LayerNorm.bias$
3. $L_2 = cls.predictions.transform.dense.bias$

The micro-tuning task is similar to its pretraining objective, the masked token task. In order to keep the tuning to few epochs with few masking combinations, and to avoid the randomness of the masking, we chose a periodic masking strategy, wherein the masking (or absence of masking) repeats at every $P$th token. Consider the masking blueprint $(k, m)$, $P = k + m$. To obtain an input from the text, we keep the first $k$ tokens of the text and mask next $m$ tokens, repeating this pattern to the end of the text. We then generate our second input by shifting the pattern by 1 token, followed by shifting 2 tokens, and so on up to $k + m - 1$ tokens. Moreover, we can create inputs from not one but several blueprints: $[(k_1, m_1), ..., (k_n, m_n)]$. For clarity, this algorithm for creating inputs is shown in Appendix A, Figure 10.

All the inputs, randomly shuffled, constitute a 'dataset' for the micro-tuning. In our evaluations we use 10 epochs micro-tuning with learning rate 0.01 and a mix of the simplest masking blueprints: $[(2, 1), (1, 1), (1, 2), (1, 3)]$, which results in a single batch of no more than 12 inputs for any text fitting into model maximal input size. See Appendix B about the processing time and the factors to reduce it.

Our ablation study, by removing a layer (Appendix C.1) and by removing a masking blueprint (Appendix C.2), shows relative importance of the layers and the blueprints.

---

[3] https://github.com/huggingface/transformers

## 3 EVALUATIONS

The goal of this section is to demonstrate that neural embeddings do capture the semantic differences between texts, and that on average the texts with similar meanings get close to each other in the neural embedding space. No embedding can be ideal for all purposes; here we attempt to observe the behavior of neural embeddings on different data and from different points of view.

We are also comparing neural embedding with the following existing sentence embeddings:

1. all-mpnet-base-v2 and all-MiniLM-L6-v2 [4] [5] [6]
2. LaBSE Feng et al. (2022) [7] [8]
3. SGPT-125M and SGPT-5.8B Muennighoff (2022) [9] [10] [11]

### 3.1 CORRELATIONS WITH SEMANTIC, LEXICAL AND SYNTACTIC SIMILARITIES

In this section we review how neural embeddings correlate with semantic, lexical and syntactic similarities of paraphrases. The separation of these three qualities is possible by utilizing the quality-controlled paraphrase generation[12], introduced in Bandel et al. (2022). We generate paraphrases with different degree of similarity to the original phrase. By changing one generation control parameter (through values $0.05, 0.10, 0.15, ..., 0.95$) and keeping two other fixed (with values selected from $0.1, 0.3, 0.5, 0.7, 0.9$), we vary one specific similarity (semantic or lexical or syntactic), while keeping two others fixed. As our original phrases we take first sentences from the first 100 pairs of phrases from MRPC train dataset - Microsoft Research Paraphrase Corpus Dolan & Brockett (2005)[13].

Our own embedding-based similarity score of a generated phrase is simply a scalar product of the neural embeddings of the generated phrase and of the original phrase. For clarity Appendix D contains more detail. The correlations obtained for the generated phrases are shown in Figure 2. Each cell value in the figure represents a correlation between the embedding-based score and the generation 'score' (control parameter value of the generation). For example, the top right corner of the first (left) heatmap in Figure 2 has a dark-blue (high) correlation value; in this case the semantic and syntactic similarities were kept constant at high value 0.9, while the lexical similarity has been varying.

All the correlations are positive, and are especially strong for lexical and semantic qualities. The correlations are also positive for the existing usual kinds of embeddings that we selected for comparison. Since all the correlations are positive, we can visualize and compare the goodness of neural embeddings vs usual embeddings by the ratio of neural embeddings correlations to the usual embeddings correlations. Figures 3, 4, 5, 6 and 7 show such comparisons with several known embeddings, in scale intentionally cut to the same interval (0.5, 1.5).

Accordingly to these figures, the embeddings-based scores that use neural embeddings correlate better with semantic similarity and with lexical similarity. Other embeddings work better for syntactic similarity. Stronger correlation with semantic similarity is unambiguously good for an embedding. Lexical and syntactic similarities are supposedly (at least approximately) untangled from semantics, so a stronger correlation with them may be considered as a desired or not desired feature, depending on downstream goals.

MRPC sentences are typically long. In appendix E we repeat our evaluation on short sentences, observing even better performance of neural embeddings in comparison to the other embeddings.

---

[4] https://www.sbert.net/docs/pretrained_models.html

[5] https://huggingface.co/sentence-transformers/all-mpnet-base-v2

[6] https://huggingface.co/sentence-transformers/all-MiniLM-L6-v2

[7] https://tfhub.dev/google/LaBSE/2

[8] https://huggingface.co/sentence-transformers/LaBSE

[9] https://github.com/Muennighoff/sgpt

[10] https://huggingface.co/Muennighoff/SGPT-125M-weightedmean-nli-bitfit

[11] https://huggingface.co/Muennighoff/SGPT-5.8B-weightedmean-nli-bitfit

[12] https://github.com/IBM/quality-controlled-paraphrase-generation

[13] https://huggingface.co/datasets/glue/viewer/mrpc/train

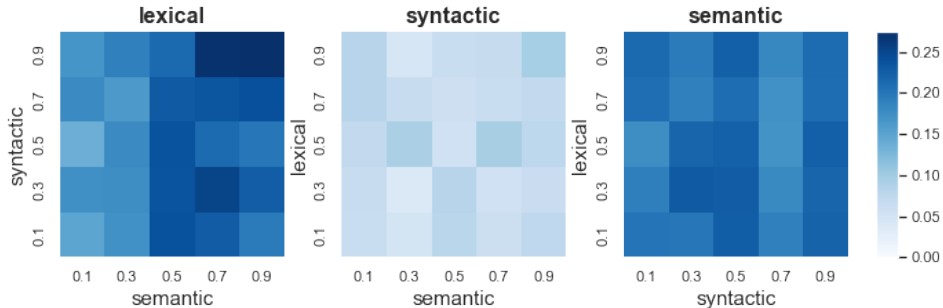

Figure 2: Evaluation of neural embeddings on generated phrases with controlled similarity. The original phrases are from MRPC. Each cell represents Kendall Tau (variant c) correlation of the embedding-based score with a selected kind of similarity: lexical similarity on the left heatmap, syntactic similarity on the middle heatmap, and semantic similarity on the right heatmap. The X and Y axes show the similarity by the fixed qualities.

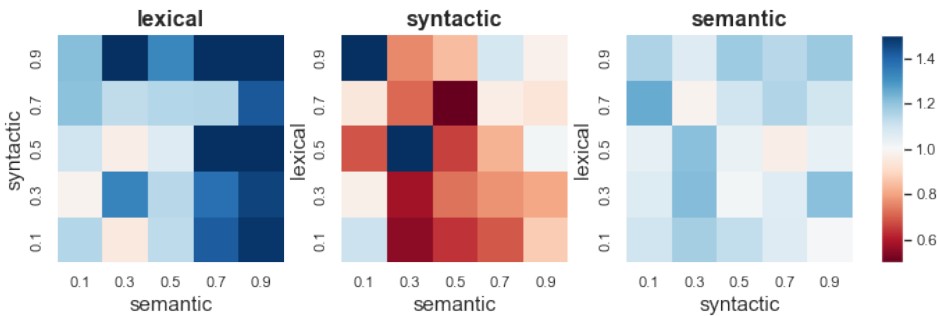

Figure 3: Ratio of correlations of neural embeddings to correlations of all-mpnet-base-v2 embeddings.

## 3.2 TRIPLETS ANCHOR-POSITIVE-NEGATIVE

If we compare the semantic similarity between texts $B$ and $C$ with regard to text $A$, and if $B$ is semantically closer to $A$, then we would want the scalar product of the embeddings of $B$ and $A$ to be higher than the product of the embeddings of $C$ and $A$:

$$sim(A, B) > sim(A, C) \iff (E_A * E_B) > (E_A * E_C) \tag{2}$$

In this section we will take triplets of texts with the similarity $sim$ satisfying $sim(A, B) > sim(A, C)$ from several datasets, and we will count the fraction of triplets for which Equation 2 is not satisfied.

We can get the triplets $(A, B, C)$ from any dataset of texts grouped by semantic similarity, so that semantically similar texts belong to the same group. Such selection of triplets was used for grouped images in Wang et al. (2014); Hoffer & Ailon (2015) and in related computer vision works. Selecting 'anchor' $A$ and 'positive' $B$ from the same group, and 'negative' $C$ from any other group provides a test: We should verify $(E_A * E_B) > (E_A * E_C)$. With a dataset in hand, we create all possible triplets and then count the fraction of the triplets for which this condition is not satisfied. The datasets we use, listed with the short notations of Tables 1 and 2:

1. *mrpc*: Composed of all sentence pairs that were annotated as similar in the 'train' dataset of MRPC - Microsoft Research Paraphrase Corpus Dolan & Brockett (2005)[14]. There are 2474 such pairs, hence we have 2474 groups, each group consisting of two texts (sentences).
2. *sts*: Composed of all sentence pairs that were annotated with similarity score at least 4 (on the scale 1-5) in STS 'test' (Englsh language) dataset (Cer et al., 2017; May, 2021)[15]. There are 338 such pairs, hence we jave 338 group, each group consisting of two texts (sentences).

---

[14]https://huggingface.co/datasets/glue/viewer/mrpc/train
[15]https://huggingface.co/datasets/stsb_multi_mt

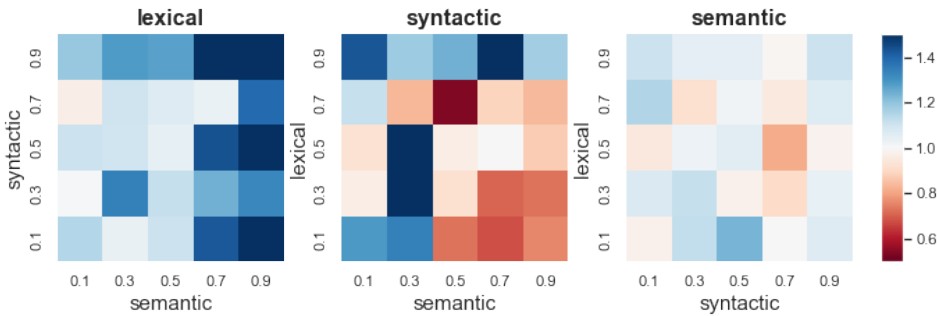

Figure 4: Ratio of correlations of neural embeddings to correlations of all-MiniLM-L6-v2 embeddings.

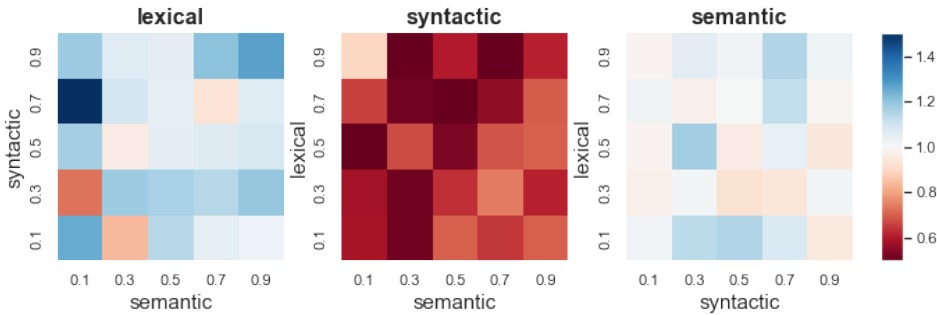

Figure 5: Ratio of correlations of neural embeddings to correlations of LaBSE embeddings.

3. *catasaurus*: First 5000 pairs of paraphrases from the dataset catasaurus_paraphrase_dataset2[16]

4. *asset/t*: Subset 'test' of the dataset asset[17] Alva-Manchego et al. (2020) - the dataset of simplified sentences.

5. *cestwc/t*: 5000 samples from the subset 'tapaco' of the dataset cestwc/paraphrase[18] dataset. The samples are selected as first 500 classes (by labels), with 10 first texts selected for each class.

6. *cestwc/q*: Composed as *cestwc/t*, but using the subset 'quora'.

7. *cestwc/c*: Composed as *cestwc/t*, but using the subset 'coco'.

8. *summ-g*: 1700 generated summaries of the dataset SummEval[19] Fabbri et al. (2021). Each summary is labeled by the document for which it was generated; there are 100 documents, 17 generated summaries for each document.

9. *summ-r*: 1100 human-written summaries of the dataset SummEval. Each summary is labeled by the document for which it was generated; there are 11 reference summaries for each document.

10. *text-summ-g*: This dataset composed by adding the texts of the documents to the dataset *summ-g*, but with restriction that only the document texts serve as $A$, and only the summaries serve as $B$ and $C$.

11. *text-summ-r*: Composed as *text-summ-g*, but with human-written reference summaries instead of the generated summaries.

The above datasets are diverse: besides paraphrased sentences, there are paraphrased questions (cestwc/quora), sentence simplifications (asset), related summaries created from the same text (summ-g and summ-r), and related text-summary pairs (text-summ-g and text-summ-r). In order to have a

---

[16]https://huggingface.co/datasets/catasaurus/paraphrase-dataset2

[17]https://github.com/facebookresearch/asset

[18]https://huggingface.co/datasets/cestwc/paraphrase

[19]https://github.com/Yale-LILY/SummEval

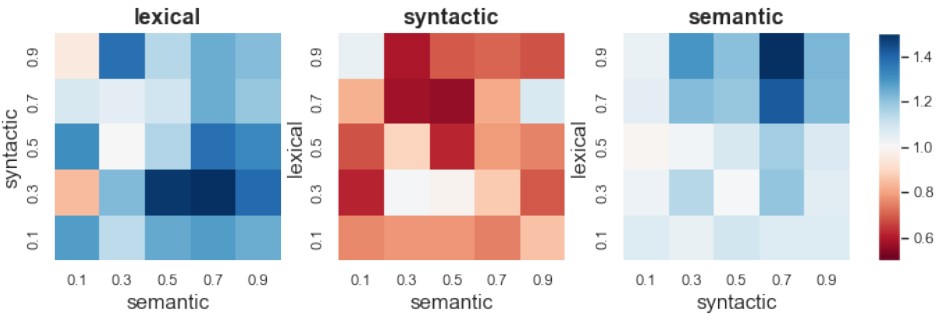

Figure 6: Ratio of correlations of neural embeddings to correlations of SGPT-125M embeddings.

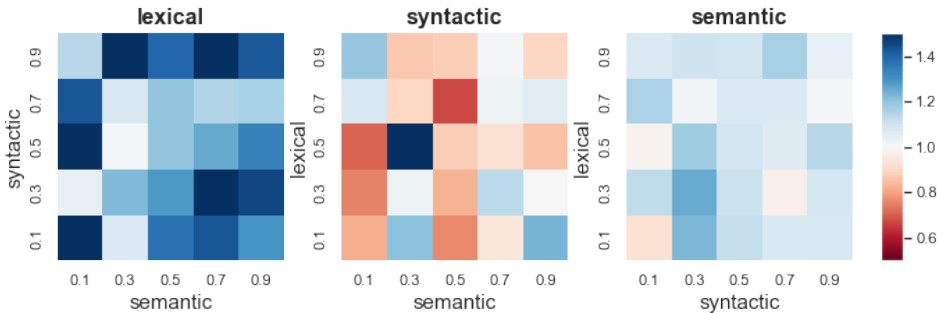

Figure 7: Ratio of correlations of neural embeddings to correlations of SGPT-5.8B embeddings.

manageable numbers of triplets, for some known datasets we took subsets or only parts of them, as specified above.

Tables 1 and 2 contain the results of our evaluation. The result for 'text-summ-g' is uniquely sensitive to random seed and the count $wrong = 44$ can shift up to 10 units up or down.

Here we considered normalized embeddings (or, equivalently, cosine as a measure of similarity). Having unnormalized embeddings from the models all-mpnet-base-v2, all-MiniLM-L6-v2 and LaBSE does not make a meaningful difference (Appendix F).

The *embedding* column shows the kind of embeddings used. Neural embeddings are obtained using the bert-base-uncased model, the layers listed in Section 2 and micro-tuning with the blueprints $[(2,1),(1,1),(1,2),(1,3)]$, 10 epochs and learning rate 0.01, as described in Section 2.

The *error* column shows the fraction of 'broken' triplets $(A, B, C)$ - the triplets for which Eq.2 is not satisfied. The *intersect* column $I$ shows the fraction of the broken triplets that the considered embeddings have in common with neural embeddings:

$$I = \frac{T_{emb} \bigcap T_{neural}}{min(|T_{emb}|, |T_{neural}|)} \qquad (3)$$

Here $I$ is the intersect, $T_{emb}$ is a set of the triplets broken by the considered embeddings, and $T_{neural}$ is a set of the triplets broken by the neural embeddings. The intersect is not listed for the neural embeddings, for which it equals 1.

The *same* column shows the average value of the scalar product $(E_A * E_B)$ over all the triplets, i.e. the average value of the product between embeddings of the texts which were selected as semantically close (same group). Similarly, the *diff* column shows the average value of the scalar product $(E_A * E_C)$ over all the triplets, i.e. the average value of the product between embeddings of the texts which were selected as semantically different (different groups).

In most datasets the neural embeddings and the known embeddings behave differently. Their errors are largely disjoint, with the overlap $I$ below $50\%$ for many datasets in Tables 1 and 2. Another

Table 1: Performance of embeddings, measured as the ability to conform to the $ABC$-relation by Eq.2. Column *error* is the fraction of broken $ABC$-relations. Column *total* is the number of the triplets, and column *wrong* is the number of broken triplets; $error = wrong/total$. $I$ is the intersection with $Neural$ errors, by Eq.3. Column *same* is the average value of embeddings product for semantically similar $AB$ texts; column *diff* is the product for semantically different $AC$. The best and the next-best embeddings are marked cyan and green correspondingly.

| dataset | embedding | error | total | wrong | $I$ | same | diff |
|---|---|---|---|---|---|---|---|
| mrpc | Neural | 6.1e-5 | 24472808 | 1491 | | 0.54 | 0.02 |
| | all-mpnet-base-v2 | 6.6e-5 | | 1621 | 0.72 | 0.85 | 0.06 |
| | all-MiniLM-L6-v2 | 8.3e-5 | | 2024 | 0.78 | 0.83 | 0.06 |
| | LaBSE | 6.4e-5 | | 1574 | 0.74 | 0.83 | 0.17 |
| | SGPT-125M | 7.9e-5 | | 1945 | 0.78 | 0.86 | 0.18 |
| | SGPT-5.8B | 5.6e-5 | | 1365 | 0.77 | 0.87 | 0.13 |
| sts | Neural | 1.0e-3 | 455624 | 455 | | 0.53 | 0.03 |
| | all-mpnet-base-v2 | 1.8e-3 | | 815 | 0.64 | 0.84 | 0.03 |
| | all-MiniLM-L6-v2 | 2.1e-3 | | 945 | 0.67 | 0.84 | 0.03 |
| | LaBSE | 1.0e-3 | | 462 | 0.69 | 0.85 | 0.15 |
| | SGPT-125M | 1.0e-3 | | 460 | 0.69 | 0.85 | 0.09 |
| | SGPT-5.8B | 9.1e-4 | | 414 | 0.71 | 0.86 | 0.06 |
| catasaurus | Neural | 7.0e-5 | 99980000 | 6998 | | 0.79 | 0.02 |
| | all-mpnet-base-v2 | 6.3e-5 | | 6325 | 0.88 | 0.97 | 0.05 |
| | all-MiniLM-L6-v2 | 6.4e-5 | | 6356 | 0.89 | 0.97 | 0.04 |
| | LaBSE | 6.7e-5 | | 6655 | 0.88 | 0.98 | 0.21 |
| | SGPT-125M | 6.3e-5 | | 6308 | 0.88 | 0.97 | 0.19 |
| | SGPT-5.8B | 6.1e-5 | | 6083 | 0.89 | 0.97 | 0.15 |
| asset/t | Neural | 4.8e-4 | 155511620 | 74731 | | 0.51 | 0.01 |
| | all-mpnet-base-v2 | 1.8e-5 | | 2838 | 0.05 | 0.87 | 0.04 |
| | all-MiniLM-L6-v2 | 1.4e-5 | | 2110 | 0.44 | 0.87 | 0.03 |
| | LaBSE | 1.3e-4 | | 20305 | 0.41 | 0.84 | 0.14 |
| | SGPT-125M | 9.7e-5 | | 15028 | 0.31 | 0.86 | 0.13 |
| | SGPT-5.8B | 2.0e-5 | | 3051 | 0.57 | 0.88 | 0.10 |
| cestwc/t | Neural | 4.2e-2 | 224550000 | 9509234 | | 0.31 | 0.03 |
| | all-mpnet-base-v2 | 1.6e-2 | | 3683569 | 0.37 | 0.64 | 0.12 |
| | all-MiniLM-L6-v2 | 2.1e-2 | | 4607873 | 0.43 | 0.63 | 0.14 |
| | LaBSE | 1.3e-2 | | 2972922 | 0.54 | 0.76 | 0.22 |
| | SGPT-125M | 1.9e-2 | | 4282170 | 0.42 | 0.68 | 0.15 |
| | SGPT-5.8B | 9.0e-3 | | 2011732 | 0.46 | 0.69 | 0.10 |
| cestwc/q | Neural | 1.0e-2 | 224550000 | 2280159 | | 0.38 | 0.07 |
| | all-mpnet-base-v2 | 3.1e-4 | | 69480 | 0.62 | 0.81 | 0.06 |
| | all-MiniLM-L6-v2 | 4.4e-4 | | 98295 | 0.65 | 0.79 | 0.06 |
| | LaBSE | 2.9e-3 | | 648334 | 0.44 | 0.75 | 0.30 |
| | SGPT-125M | 2.1e-3 | | 479279 | 0.43 | 0.77 | 0.22 |
| | SGPT-5.8B | 9.2e-4 | | 205408 | 0.53 | 0.79 | 0.19 |
| cestwc/c | Neural | 1.1e-1 | 224550000 | 24563942 | | 0.23 | 0.08 |
| | all-mpnet-base-v2 | 3.6e-2 | | 8085033 | 0.60 | 0.56 | 0.10 |
| | all-MiniLM-L6-v2 | 3.8e-2 | | 8500814 | 0.61 | 0.55 | 0.09 |
| | LaBSE | 8.3e-2 | | 18588451 | 0.53 | 0.52 | 0.27 |
| | SGPT-125M | 4.4e-2 | | 9920519 | 0.57 | 0.60 | 0.17 |
| | SGPT-5.8B | 3.3e-2 | | 7300297 | 0.61 | 0.62 | 0.15 |

difference between neural and other embeddings is how they spread out in the embedding space. The average product of embeddings is higher for all known embeddings than for neural embeddings. This is true both for semantically similar and for semantically different texts - see the *same* and *diff* columns.

Table 2: Performance of embeddings, as the ability to conform to the $ABC$-relation by Eq.2.

| dataset | embedding | error | total | wrong | $I$ | same | diff |
|---|---|---|---|---|---|---|---|
| text-summ-g | Neural | 1.5e-5 | 2861100 | 44 | | 0.42 | 0.03 |
| | all-mpnet-base-v2 | 3.7e-5 | | 107 | 0.04 | 0.83 | 0.11 |
| | all-MiniLM-L6-v2 | 1.1e-4 | | 321 | 0.06 | 0.81 | 0.11 |
| | LaBSE | 2.7e-3 | | 7803 | 0.55 | 0.75 | 0.29 |
| | SGPT-125M | 8.0e-3 | | 22771 | 0.47 | 0.70 | 0.22 |
| | SGPT-5.8B | 8.7e-4 | | 2495 | 0.34 | 0.77 | 0.20 |
| text-summ-r | Neural | 6.7e-3 | 1197900 | 7988 | 0.16 | 0.27 | 0.03 |
| | all-mpnet-base-v2 | 2.3e-3 | | 2779 | 0.27 | 0.73 | 0.10 |
| | all-MiniLM-L6-v2 | 2.6e-3 | | 3075 | 0.27 | 0.69 | 0.10 |
| | LaBSE | 2.3e-2 | | 27884 | 0.46 | 0.62 | 0.22 |
| | SGPT-125M | 1.1e-2 | | 12967 | 0.16 | 0.71 | 0.21 |
| | SGPT-5.8B | 4.2e-3 | | 4987 | 0.16 | 0.72 | 0.17 |
| summ-g | Neural | 4.4e-3 | 45777600 | 203275 | 0.20 | 0.45 | 0.03 |
| | all-mpnet-base-v2 | 2.9e-4 | | 13330 | 0.31 | 0.82 | 0.10 |
| | all-MiniLM-L6-v2 | 6.3e-4 | | 28986 | 0.31 | 0.81 | 0.10 |
| | LaBSE | 2.3e-3 | | 106435 | 0.27 | 0.78 | 0.24 |
| | SGPT-125M | 6.3e-3 | | 287977 | 0.20 | 0.78 | 0.25 |
| | SGPT-5.8B | 1.5e-3 | | 70610 | 0.22 | 0.80 | 0.19 |
| summ-r | Neural | 3.7e-2 | 11979000 | 440827 | 0.34 | 0.22 | 0.02 |
| | all-mpnet-base-v2 | 6.4e-3 | | 76309 | 0.45 | 0.67 | 0.09 |
| | all-MiniLM-L6-v2 | 8.1e-3 | | 97028 | 0.44 | 0.64 | 0.09 |
| | LaBSE | 1.8e-2 | | 211836 | 0.39 | 0.61 | 0.18 |
| | SGPT-125M | 1.6e-2 | | 190809 | 0.34 | 0.66 | 0.18 |
| | SGPT-5.8B | 7.1e-3 | | 85433 | 0.44 | 0.67 | 0.15 |

## 3.3 AGREEMENT WITH POSITIVELY AND NEGATIVELY ANNOTATED PAIRS

The datasets containing the pairs of sentences deemed not similar provide another way of evaluation: any similar pair must have higher product of embeddings than any non-similar pair:

$$sim(A, B) > sim(C, D) \iff (E_A * E_B) > (E_C * E_D) \tag{4}$$

Table 3 contains the results of this evaluation.

Table 3: Performance of embeddings, measured as the ability to grade a pair of sentences labeled similar higher than a pair of sentences labeled as non-similar. Column $total$ is the total number of tuples (similar pair, non-similar pair). Column $wrong$ is the number of errors, so that $error = wrong/total$. Column *same* is the average value of embeddings product for pairs labeled as similar; column *diff* is the average product for pairs labeled as different.

| dataset | embedding | error | total | wrong | $I$ | same | diff |
|---|---|---|---|---|---|---|---|
| mrpc | Neural | 2.6e-1 | 2953956 | 772375 | 0.62 | 0.54 | 0.42 |
| | all-mpnet-base-v2 | 2.2e-1 | | 651299 | 0.58 | 0.85 | 0.71 |
| | all-MiniLM-L6-v2 | 2.5e-1 | | 736308 | 0.60 | 0.83 | 0.71 |
| | LaBSE | 2.2e-1 | | 637497 | 0.67 | 0.83 | 0.70 |
| | SGPT-125M | 2.4e-1 | | 695606 | 0.62 | 0.86 | 0.76 |
| | SGPT-5.8B | 2.1e-1 | | 607056 | 0.61 | 0.87 | 0.75 |
| sts | Neural | 9.9e-2 | 180492 | 17893 | 0.56 | 0.53 | 0.29 |
| | all-mpnet-base-v2 | 3.3e-2 | | 5889 | 0.52 | 0.84 | 0.33 |
| | all-MiniLM-L6-v2 | 4.0e-2 | | 7216 | 0.61 | 0.84 | 0.37 |
| | LaBSE | 6.6e-2 | | 11948 | 0.50 | 0.85 | 0.56 |
| | SGPT-125M | 4.1e-2 | | 7333 | 0.56 | 0.85 | 0.48 |
| | SGPT-5.8B | 2.0e-2 | | 3626 | 0.59 | 0.86 | 0.42 |

MRPC 'train' dataset contains 2474 similar and 1194 non-similar pairs. The STS 'test' dataset contains 374 pairs with score 4 or 5 - which we used as similar, and 479 pairs with score 2 or 1 - which we used as non-similar.

### 3.4 CORRELATIONS WITH HUMAN SCORES

If dataset contains human-scored pairs of texts or sentences, with the score reflecting at least in some sense a similarity between the texts, then the product of embeddings should correlate with the score. In Table 4 we show examples of such correlations.

Table 4: Correlations of embedding product with human scores.

| dataset | score | embedding | kendall-c | kendall-b | spearman |
|---|---|---|---|---|---|
| sts | similarity | Neural | 0.478 | 0.482 | 0.658 |
| | | all-mpnet-base-v2 | 0.651 | 0.655 | 0.834 |
| | | all-MiniLM-L6-v2 | 0.638 | 0.642 | 0.820 |
| | | LaBSE | 0.537 | 0.541 | 0.723 |
| | | SGPT-125M | 0.608 | 0.612 | 0.795 |
| | | SGPT-5.8B | 0.675 | 0.680 | 0.857 |
| summeval | coherence | Neural | 0.100 | 0.097 | 0.137 |
| | | all-mpnet-base-v2 | 0.115 | 0.112 | 0.159 |
| | | all-MiniLM-L6-v2 | 0.096 | 0.093 | 0.134 |
| | | LaBSE | 0.113 | 0.110 | 0.154 |
| | | SGPT-125M | 0.156 | 0.151 | 0.216 |
| | | SGPT-5.8B | 0.169 | 0.164 | 0.232 |
| summeval | consistency | Neural | 0.098 | 0.157 | 0.200 |
| | | all-mpnet-base-v2 | 0.075 | 0.120 | 0.153 |
| | | all-MiniLM-L6-v2 | 0.087 | 0.138 | 0.176 |
| | | LaBSE | 0.077 | 0.122 | 0.156 |
| | | SGPT-125M | 0.011 | 0.017 | 0.022 |
| | | SGPT-5.8B | 0.068 | 0.108 | 0.137 |
| summeval | fluency | Neural | 0.063 | 0.084 | 0.109 |
| | | all-mpnet-base-v2 | 0.056 | 0.075 | 0.097 |
| | | all-MiniLM-L6-v2 | 0.064 | 0.086 | 0.112 |
| | | LaBSE | 0.055 | 0.074 | 0.095 |
| | | SGPT-125M | -0.004 | -0.006 | -0.007 |
| | | SGPT-5.8B | 0.023 | 0.030 | 0.039 |
| summeval | relevance | Neural | 0.190 | 0.188 | 0.263 |
| | | all-mpnet-base-v2 | 0.200 | 0.198 | 0.278 |
| | | all-MiniLM-L6-v2 | 0.169 | 0.167 | 0.236 |
| | | LaBSE | 0.217 | 0.215 | 0.301 |
| | | SGPT-125M | 0.198 | 0.196 | 0.275 |
| | | SGPT-5.8B | 0.230 | 0.228 | 0.318 |

The 'sts' (STS test dataset) is scored by similarity of the phrases. The 'summeval' (SummEval dataset Fabbri et al. (2021)) has human-scored, by 3 expert, each of 1700 summaries, which were generated by 17 models for 100 documents. Each summary is scored for coherence, consistency, fluency and relevance (we are using the score averaged over the 3 expert scores). Neural embeddings are the best in correlations with consistency, which is a quality that is probably the closest to semantics.

## 4 CONCLUSION

In this paper we introduced *neural embeddings*, a new method for representing text as vectors. To our knowledge, this is the first attempt to obtain representations in this manner. In Appendix G we listed the factors by which the construction of neural embeddings differs from the usual embeddings; based on the presented there observations we conclude that micro-tuning is the most important factor.

Our observations in Section 3.1 and in Appendix E show that neural embeddings make more emphasis on semantic and lexical qualities, and less on syntactic qualities. We also observed through Section 3 that neural embeddings behave distinctly differently from all the embeddings we included in our comparisons. As an illustration of the difference, in Appendix H we show results for a simple 'ensemble': the concatenation of neural and SGPT-125M embeddings. As expected, in many cases the concatenation outperforms them both.

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

## A    Neural embeddings and micro-tuning

In section 2 we defined neural embedding for text. For clarity, the pseudocode is shown in Figure 8.

---

Given:
    Model $M$
    Weights $\{W_j\}$ from selected layers $\{L_j\}$ of $M$, all other layers are always frozen
Obtain embedding for text $T$:
Split $T = T_0 + T_1 + ...$
    Each text chunk $T_i$ consists of max number of sentences fitting input size of the model $M$
Set model layers $\{L_j\}$ weights to $\{W_j\}$
Initialize $inputs\_and\_labels$
**for each** chunk $T_i$:
    Make inputs and labels
    Add inputs and labels to $inputs\_and\_labels$
Tune model $M$ on $inputs\_and\_labels$
Take the tuned weights $\{W_j'\}$ of the layers $\{L_j\}$.
Calculate changes with respect to original weights: $dW_j = W_j' - W_j$
Normalize and concatenate them: $E_c = \|_{j=1}^{m} dW_j'/|dW_j|$
Normalize the result: $E = E_c/|E_c|$
**return** $E$

---

Figure 8: Producing neural embedding for text $T$. Notice that the weights of layers $\{L_j\}$ are always reset to the original values before obtaining neural embedding for a text. Here the symbol $\|$ means concatenation, e.g. $\|_{j=1}^{m} a_j$ is a concatenation of $a_0, a_1, ... , a_m$.

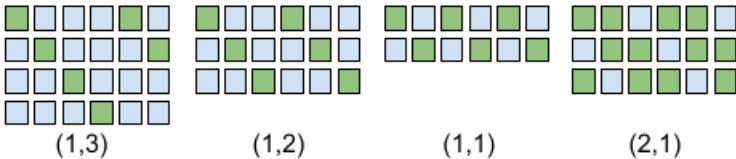

Figure 9: The inputs for micro-tuning on text with length 6 tokens, obtained by the blueprints $[(2,1),(1,1),(1,2),(1,3)]$. The tokens kept (not masked) are colored green. Labels are made for predicting the masked tokens. Altogether there are 12 inputs, represented here by 12 rows of length 6.

Our choice of layers for the evaluations presented in Section 3 is influenced by several factors. Practically, any 2D layers (for example attention weights) are too large to create an embedding. The decoder bias layer ('cls.predictions.decoder.bias') is 1D but also too large, with a size 30522. It is reasonable to chose layers from top of the model, when most of the processing is done by the bulk of the transformer layers, and the patterns emerged; this means that the last transformer block and the classification layer are the good places to pick the layers from. In terms of huggingface BERT, it is the layers 'bert.encoder.layer.11' and 'cls.predictions'. Empirically, we found that contiguous set of two or three layers at the end of these blocks works the best, e.g.

1. $L_0 = cls.predictions.transform.LayerNorm.weight$
2. $L_1 = cls.predictions.transform.LayerNorm.bias$
3. $L_2 = cls.predictions.transform.dense.bias$

or

1. $L_0 = bert.encoder.layer.11.output.LayerNorm.weight$
2. $L_1 = bert.encoder.layer.11.output.LayerNorm.bias$

3. $L_2 = bert.encoder.layer.11.output.dense.bias$

Taking the same layers from lower blocks worsen the neural embeddings performance. In our evaluations we used the first choice; in Appendix C.1 we present the ablation results with one of the layers excluded.

Figure 9 is an illustration for obtaining the inputs for text of 6 tokens with the blueprints $[(2, 1), (1, 1), (1, 2), (1, 3)]$. There are only 12 inputs for any text fitting into model maximal input size - or even less than 12 inputs if the text length is less than 4 tokens. This fits into single batch. For clarity, the pseudocode for obtaining inputs for micro-tuning is shown in Figure 10. In Appendix C.2 we present the ablation results with one of the blueprints excluded.

Interestingly, if we do not mask tokens at all (but require prediction for all tokens), we still get reasonable embeddings, albeit with much lower 'quality' (if evaluated by any criteria of Section 3). The loss is still defined by cross entropy, no matter how easy the prediction task is, and for the same reason there is always weights difference in Equation 1.

---

Given:
    Blueprints for masking: $[(k_1, m_1), ..., (k_n, m_n)]$
    Text chunk tokens $T = [t_0, t_1, ...]$
Initialize empty list $inputs\_and\_labels$
**for each** masking blueprint $(k_i, m_i)$:
    Period $P = k_i + m_i$
    Maximal shift $S_{max} = \min(P, len(T))$
    **for** $s$ in range($S_{max}$):
        input $I$ = deepcopy($T$)
        initialize labels $L$, with length $len(L) = len(T)$
        **for** $j$ in range($len(T)$):
            offset $R = (j - s)\%P$
            **if** $R >= k_i$**:**
                set label: $L[j] = I[j]$
                mask token: $I[j] = mask$
        Append duple $(I, L)$ to $inputs\_and\_labels$
**return** $inputs\_and\_labels$

---

Figure 10: Creating inputs and labels from a chunk of text. The given length of the chunk $len(T)$ must not exceed the max allowed size of model input.

## B  FASTER MICRO-TUNING

For convenience or reproducibility, the neural embeddings can be obtained simply by using the BERT model as it is, with all the layers frozen, except the selected layers, as described in Section 2. But there are several simplifications that make obtaining neural embeddings almost as fast as a simple forward run of BERT.

Since the micro-tuning 'dataset' is extremely small, all the inputs can be kept on GPU through all epochs. More importantly, since all the selected (for embeddings) layers are located at the top of the model, there is no need to repeat the forward run through all the lower transformer blocks. The hidden states from the last block preceding the layers of interest can be obtained only once, at the first epoch, and then reused in all the next epochs. In result, at each epoch except the first one, the micro-tuning deals only with the top of the model (separated from the rest). Again, since the micro-tuning 'dataset' is extremely small, even these hidden states can be kept on GPU, unless we are dealing with a really long text of too many input windows.

For example, using simply the whole BERT model (with 10 epochs), the average time of obtaining neural embedding on GPU P100 for MRPC sentence is 0.24 sec., and for STS sentence is 0.16 sec. With hidden states reused, these times become 0.096 sec. and 0.057 sec. Reducing the number of epochs offers another option: Switching from 10 to 5 epochs further decreases these times almost twice (with evaluation results in most cases worse only but several percents). However, this is still

much slower than a time around 0.006 - 0.008 sec. by 'all-mpnet-base-v2', 'LaBSE' or 'SGPT-145M' (both for MRPC and STS).

## C ABLATION: REMOVING LAYERS AND MASK-BLUEPRINTS

### C.1 ABLATION: LAYERS

In this section we present ablation study results for excluding one of the layers from using in constructing the neural embedding (three layers were used in our evaluations, Section 3). We are using the same micro-tuning as in Section 3, but with 5 (not 10) epochs. The Neural embedding is compared in Tables 5, 6, 7 and 8 with the embedding versions Neural-L0, Neural-L1 and Neural-L2, where the suffix indicates the layer that was not used. Reminder, the layers are:

1. $L_0 = cls.predictions.transform.LayerNorm.weight$
2. $L_1 = cls.predictions.transform.LayerNorm.bias$
3. $L_2 = cls.predictions.transform.dense.bias$

Table 5: Ablation: removal of a layer. Triplets.

| dataset | embedding | error | total | wrong | same | diff |
|---|---|---|---|---|---|---|
| mrpc | Neural | 7.2e-5 | 24472808 | 1757 | 0.53 | 0.03 |
| | Neural-L0 | 1.2e-4 | | 2996 | 0.55 | 0.01 |
| | Neural-L1 | 7.0e-5 | | 1713 | 0.51 | 0.03 |
| | Neural-L2 | 1.6e-4 | | 3905 | 0.52 | 0.04 |
| sts | Neural | 1.1e-3 | 455624 | 516 | 0.53 | 0.04 |
| | Neural-L0 | 1.2e-3 | | 541 | 0.56 | 0.03 |
| | Neural-L1 | 1.1e-3 | | 517 | 0.51 | 0.04 |
| | Neural-L2 | 1.5e-3 | | 697 | 0.51 | 0.05 |
| catasaurus | Neural | 7.1e-5 | 99980000 | 7090 | 0.77 | 0.02 |
| | Neural-L0 | 7.4e-5 | | 7445 | 0.81 | 0.01 |
| | Neural-L1 | 7.0e-5 | | 7037 | 0.75 | 0.02 |
| | Neural-L2 | 7.1e-5 | | 7123 | 0.75 | 0.03 |
| asset/t | Neural | 6.4e-4 | 155511620 | 99951 | 0.50 | 0.02 |
| | Neural-L0 | 6.2e-4 | | 96207 | 0.52 | 0.01 |
| | Neural-L1 | 8.2e-4 | | 128230 | 0.48 | 0.02 |
| | Neural-L2 | 1.3e-3 | | 196091 | 0.48 | 0.03 |
| cestwc/t | Neural | 4.4e-2 | 224550000 | 9991006 | 0.30 | 0.03 |
| | Neural-L0 | 4.6e-2 | | 10253234 | 0.33 | 0.03 |
| | Neural-L1 | 4.6e-2 | | 10353332 | 0.29 | 0.03 |
| | Neural-L2 | 4.9e-2 | | 10969280 | 0.29 | 0.03 |
| cestwc/q | Neural | 1.2e-2 | 224550000 | 2645701 | 0.38 | 0.08 |
| | Neural-L0 | 1.2e-2 | | 2584319 | 0.41 | 0.08 |
| | Neural-L1 | 1.3e-2 | | 2835001 | 0.36 | 0.07 |
| | Neural-L2 | 1.7e-2 | | 3711619 | 0.37 | 0.09 |
| cestwc/c | Neural | 1.2e-1 | 224550000 | 25984217 | 0.23 | 0.09 |
| | Neural-L0 | 1.1e-1 | | 25249882 | 0.26 | 0.09 |
| | Neural-L1 | 1.2e-1 | | 27228191 | 0.22 | 0.08 |
| | Neural-L2 | 1.3e-1 | | 30012156 | 0.23 | 0.09 |

From Tables 5, 6, 7 and 8 we can conclude that the layer L2 is the most important: the embeddings Neural-L2 never done well in the tables. The layer L0 is the least important and maybe can be excluded, depending on a task: the embeddings Neural-L0 done well in many cases in the tables.

### C.2 ABLATION: MASK-BLUEPRINTS

Here in Tables 9, 10, 11 and 12 we present results of ablation by excluding masking blueprints from construction of neural embedding. Similar to the ablation study of layers, we compare here Neural embeddings with embedding versions Neural-B0, Neural-B1, Neural-B2 and Neural-B3, where the

Table 6: Ablation: removal of a layer. Triplets. Texts and summaries.

| dataset | embedding | error | total | wrong | same | diff |
|---|---|---|---|---|---|---|
| text-summ-g | neural | 3.8e-5 | 2861100 | 109 | 0.42 | 0.03 |
| | neural-L0 | 3.1e-4 | | 897 | 0.43 | 0.04 |
| | neural-L1 | 7.7e-5 | | 221 | 0.39 | 0.03 |
| | neural-L2 | 3.7e-4 | | 1067 | 0.38 | 0.04 |
| text-summ-r | neural | 6.9e-3 | 1197900 | 8277 | 0.26 | 0.03 |
| | neural-L0 | 9.3e-3 | | 11167 | 0.28 | 0.03 |
| | neural-L1 | 9.3e-3 | | 11184 | 0.24 | 0.02 |
| | neural-L2 | 1.5e-2 | | 18101 | 0.24 | 0.03 |
| summ-g | neural | 5.6e-3 | 45777600 | 255665 | 0.44 | 0.03 |
| | neural-L0 | 6.9e-3 | | 317828 | 0.45 | 0.03 |
| | neural-L1 | 6.6e-3 | | 300512 | 0.42 | 0.03 |
| | neural-L2 | 8.2e-3 | | 376727 | 0.43 | 0.04 |
| summ-r | neural | 3.6e-2 | 11979000 | 435033 | 0.22 | 0.03 |
| | neural-L0 | 4.5e-2 | | 540283 | 0.24 | 0.02 |
| | neural-L1 | 4.7e-2 | | 565869 | 0.21 | 0.03 |
| | neural-L2 | 5.4e-2 | | 652541 | 0.21 | 0.04 |

Table 7: Ablation: removal of a layer. Pairs similar and not similar.

| dataset | embedding | error | total | wrong | same | diff |
|---|---|---|---|---|---|---|
| mrpc | Neural | 2.6e-1 | 2953956 | 772375 | 0.54 | 0.42 |
| | Neural-L0 | 2.6e-1 | | 764793 | 0.55 | 0.43 |
| | Neural-L1 | 2.7e-1 | | 800874 | 0.51 | 0.40 |
| | Neural-L2 | 2.7e-1 | | 804656 | 0.52 | 0.41 |
| sts | Neural | 1.1e-2 | 180492 | 18975 | 0.53 | 0.29 |
| | Neural-L0 | 1.0e-2 | | 18742 | 0.56 | 0.31 |
| | Neural-L1 | 1.1e-2 | | 19569 | 0.51 | 0.28 |
| | Neural-L2 | 1.1e-2 | | 20485 | 0.51 | 0.29 |

Table 8: Ablation: removal of a layer. Correlations of embedding product with human scores.

| dataset | score | embedding | kendall-c | kendall-b | spearman |
|---|---|---|---|---|---|
| sts | similarity | Neural | 0.470 | 0.473 | 0.648 |
| | | Neural-L0 | 0.471 | 0.475 | 0.649 |
| | | Neural-L1 | 0.465 | 0.468 | 0.642 |
| | | Neural-L2 | 0.453 | 0.457 | 0.628 |
| summeval | coherence | Neural | 0.104 | 0.101 | 0.143 |
| | | Neural-L0 | 0.112 | 0.109 | 0.154 |
| | | Neural-L1 | 0.098 | 0.095 | 0.134 |
| | | Neural-L2 | 0.098 | 0.095 | 0.135 |
| summeval | consistency | Neural | 0.092 | 0.147 | 0.187 |
| | | Neural-L0 | 0.087 | 0.135 | 0.171 |
| | | Neural-L1 | 0.095 | 0.152 | 0.193 |
| | | Neural-L2 | 0.084 | 0.134 | 0.171 |
| summeval | fluency | Neural | 0.057 | 0.076 | 0.098 |
| | | Neural-L0 | 0.053 | 0.071 | 0.091 |
| | | Neural-L1 | 0.057 | 0.078 | 0.100 |
| | | Neural-L2 | 0.050 | 0.067 | 0.086 |
| summeval | relevance | Neural | 0.194 | 0.192 | 0.269 |
| | | Neural-L0 | 0.202 | 0.200 | 0.280 |
| | | Neural-L1 | 0.184 | 0.182 | 0.254 |
| | | Neural-L2 | 0.178 | 0.176 | 0.248 |

suffix indicates the blueprint excluded from constraction of the embedding. Reminding from Section 2, the blueprints are: $[(2,1),(1,1),(1,2),(1,3)]$.

Table 9: Ablation: removal of a mask-blueprint. Triplets.

| dataset | embedding | error | total | wrong | same | diff |
|---|---|---|---|---|---|---|
| mrpc | Neural | 7.2e-5 | 24472808 | 1757 | 0.53 | 0.03 |
| | Neural-B0 | 7.9e-5 | | 1932 | 0.53 | 0.03 |
| | Neural-B1 | 7.0e-5 | | 1713 | 0.53 | 0.03 |
| | Neural-B2 | 7.1e-5 | | 1747 | 0.52 | 0.02 |
| | Neural-B3 | 9.5e-5 | | 2334 | 0.50 | 0.01 |
| sts | Neural | 1.1e-3 | 455624 | 516 | 0.53 | 0.04 |
| | Neural-B0 | 1.2e-3 | | 534 | 0.53 | 0.04 |
| | Neural-B1 | 1.0e-3 | | 466 | 0.53 | 0.04 |
| | Neural-B2 | 1.4e-3 | | 653 | 0.51 | 0.04 |
| | Neural-B3 | 1.3e-3 | | 582 | 0.50 | 0.03 |
| catasaurus | Neural | 7.1e-5 | 99980000 | 7090 | 0.77 | 0.02 |
| | Neural-B0 | 7.1e-5 | | 7113 | 0.77 | 0.02 |
| | Neural-B1 | 7.1e-5 | | 7079 | 0.77 | 0.02 |
| | Neural-B2 | 7.2e-5 | | 7170 | 0.76 | 0.02 |
| | Neural-B3 | 7.3e-5 | | 7281 | 0.74 | 0.01 |
| asset/t | Neural | 6.4e-4 | 155511620 | 99951 | 0.50 | 0.02 |
| | Neural-B0 | 7.3e-4 | | 113772 | 0.50 | 0.02 |
| | Neural-B1 | 7.2e-4 | | 111335 | 0.50 | 0.02 |
| | Neural-B2 | 7.0e-3 | | 109574 | 0.49 | 0.01 |
| | Neural-B3 | 8.2e-4 | | 126789 | 0.47 | 0.01 |
| cestwc/t | Neural | 4.4e-2 | 224550000 | 9991006 | 0.30 | 0.03 |
| | Neural-B0 | 4.5e-2 | | 10101417 | 0.31 | 0.03 |
| | Neural-B1 | 4.5e-2 | | 10096884 | 0.31 | 0.03 |
| | Neural-B2 | 4.7e-2 | | 10530677 | 0.30 | 0.03 |
| | Neural-B3 | 5.0e-2 | | 11259016 | 0.28 | 0.02 |
| cestwc/q | Neural | 1.2e-2 | 224550000 | 2645701 | 0.38 | 0.08 |
| | Neural-B0 | 1.3e-2 | | 2855655 | 0.38 | 0.09 |
| | Neural-B1 | 1.3e-2 | | 2813956 | 0.38 | 0.09 |
| | Neural-B2 | 1.2e-2 | | 2626329 | 0.37 | 0.07 |
| | Neural-B3 | 1.3e-2 | | 2913397 | 0.35 | 0.06 |
| cestwc/c | Neural | 1.2e-1 | 224550000 | 25984217 | 0.23 | 0.09 |
| | Neural-B0 | 1.2e-1 | | 26374181 | 0.24 | 0.09 |
| | Neural-B1 | 1.2e-1 | | 26567529 | 0.24 | 0.10 |
| | Neural-B2 | 1.2e-1 | | 26059561 | 0.23 | 0.08 |
| | Neural-B3 | 1.2e-1 | | 26882127 | 0.20 | 0.06 |

The conclusion from Tables 9, 10, 11 and 12 is simpler than from ablation of the layers. The importance of the selected blueprints is not very different, excluding any of them leads to worse performance in most cases, yet the blueprint B1 maybe less important than the others. The blueprint B1 is the simplest blueprint (1,1) - one token kept and one token masked.

# D    CONTROLLED-GENERATED PHRASES

In Section 3.1 we used control-generated phrases for comparing lexical, syntactic and semantic representation of the phrases by neural embeddings. We reviewed how neural embeddings correlate with semantic, lexical and syntactic similarities of paraphrases. In this appendix we provide more detail. The separation of the three qualities (lexical, syntactic and semantic) is possible by utilizing the quality-controlled paraphrase generation[20], introduced in Bandel et al. (2022). We generated paraphrases with different degree of similarity to the original phrase. By changing one generation control parameter and keeping two other fixed, we varied one specific similarity (semantic or lexical or syntactic), while keeping two others fixed.

---

[20]https://github.com/IBM/quality-controlled-paraphrase-generation

Table 10: Ablation: removal of a mask-blueprint. Triplets. Summaries and texts.

| dataset | embedding | error | total | wrong | same | diff |
|---|---|---|---|---|---|---|
| text-summ-g | neural | 3.8e-5 | 2861100 | 109 | 0.42 | 0.03 |
| | neural-B0 | 3.3e-4 | | 934 | 0.41 | 0.04 |
| | neural-B1 | 3.4e-4 | | 971 | 0.41 | 0.05 |
| | neural-B2 | 3.3e-4 | | 944 | 0.39 | 0.04 |
| | neural-B3 | 6.7e-5 | | 193 | 0.38 | 0.01 |
| text-summ-r | neural | 6.9e-3 | 1197900 | 8277 | 0.26 | 0.03 |
| | neural-B0 | 8.0e-3 | | 9535 | 0.27 | 0.03 |
| | neural-B1 | 9.1e-3 | | 10894 | 0.27 | 0.04 |
| | neural-B2 | 9.4e-3 | | 11283 | 0.24 | 0.03 |
| | neural-B3 | 8.9e-3 | | 10602 | 0.22 | 0.01 |
| summ-g | neural | 5.6e-3 | 45777600 | 255665 | 0.44 | 0.03 |
| | neural-B0 | 5.9e-3 | | 270434 | 0.44 | 0.04 |
| | neural-B1 | 6.1e-3 | | 277742 | 0.44 | 0.04 |
| | neural-B2 | 6.2e-3 | | 283341 | 0.43 | 0.04 |
| | neural-B3 | 6.2e-3 | | 284366 | 0.41 | 0.02 |
| summ-r | neural | 3.6e-2 | 11979000 | 435033 | 0.22 | 0.03 |
| | neural-B0 | 4.1e-2 | | 493764 | 0.23 | 0.03 |
| | neural-B1 | 4.2e-2 | | 501948 | 0.23 | 0.03 |
| | neural-B2 | 4.3e-2 | | 516386 | 0.21 | 0.03 |
| | neural-B3 | 5.2e-2 | | 623027 | 0.19 | 0.01 |

Table 11: Ablation: removal of a mask-blueprint. Pairs similar and not similar.

| dataset | embedding | error | total | wrong | same | diff |
|---|---|---|---|---|---|---|
| mrpc | Neural | 2.6e-1 | 2953956 | 772375 | 0.54 | 0.42 |
| | Neural-B0 | 2.7e-1 | | 793688 | 0.52 | 0.42 |
| | Neural-B1 | 2.7e-1 | | 786711 | 0.53 | 0.42 |
| | Neural-B2 | 2.7e-1 | | 783147 | 0.52 | 0.41 |
| | Neural-B3 | 2.7e-1 | | 796018 | 0.50 | 0.39 |
| sts | Neural | 1.1e-2 | 180492 | 18975 | 0.53 | 0.29 |
| | Neural-B0 | 1.1e-2 | | 19953 | 0.53 | 0.30 |
| | Neural-B1 | 1.0e-2 | | 18760 | 0.53 | 0.30 |
| | Neural-B2 | 1.1e-1 | | 19328 | 0.51 | 0.28 |
| | Neural-B3 | 1.1e-2 | | 20024 | 0.50 | 0.26 |

Our own embedding-based similarity score of a generated phrase is simply a scalar product of the neural embeddings of the generated phrase and of the original phrase. In Section 3.1 we calculated correlations between the embedding-based similarity and the generation control parameter. As our original phrases we took first sentences from the first 100 pairs of phrases from MRPC train dataset - Microsoft Research Paraphrase Corpus Dolan & Brockett (2005)[21]. Each evaluation is defined by the following:

1. The selected quality (semantic or lexical or syntactic) for which we generate phrases with different degree of similarity, by varying the corresponding control parameter.
2. The fixed control parameters for the remaining two qualities.

We have chosen our fixed control parameters from the values $0.1, 0.3, 0.5, 0.7, 0.9$. Thus, with $5 \times 5 = 25$ choices of the fixed control parameters, and with 3 choices of selecting the quality to vary, we have $25 \times 3 = 75$ evaluations, i.e. 75 correlations to calculate. For each evaluation, we did the following:

1. For each phrase of our 100 original phrases, we generated new phrases with the selected control parameter taking the values $0.05, 0.10, 0.15, ..., 0.95$ (and with the other two control

---

[21]https://huggingface.co/datasets/glue/viewer/mrpc/train

Table 12: Ablation: removal of a mask-blueprint. Correlations of embedding product with human scores.

| dataset | score | embedding | kendall-c | kendall-b | spearman |
|---|---|---|---|---|---|
| sts | similarity | Neural | 0.470 | 0.473 | 0.648 |
| | | Neural-B0 | 0.460 | 0.463 | 0.636 |
| | | Neural-B1 | 0.467 | 0.470 | 0.645 |
| | | Neural-B2 | 0.466 | 0.469 | 0.643 |
| | | Neural-B3 | 0.468 | 0.471 | 0.644 |
| summeval | coherence | Neural | 0.104 | 0.101 | 0.143 |
| | | Neural-B0 | 0.092 | 0.089 | 0.126 |
| | | Neural-B1 | 0.092 | 0.089 | 0.126 |
| | | Neural-B2 | 0.097 | 0.094 | 0.133 |
| | | Neural-B3 | 0.095 | 0.092 | 0.130 |
| summeval | consistency | Neural | 0.092 | 0.147 | 0.187 |
| | | Neural-B0 | 0.083 | 0.132 | 0.168 |
| | | Neural-B1 | 0.082 | 0.130 | 0.166 |
| | | Neural-B2 | 0.091 | 0.144 | 0.183 |
| | | Neural-B3 | 0.102 | 0.162 | 0.206 |
| summeval | fluency | Neural | 0.057 | 0.076 | 0.098 |
| | | Neural-B0 | 0.047 | 0.063 | 0.081 |
| | | Neural-B1 | 0.049 | 0.066 | 0.085 |
| | | Neural-B2 | 0.057 | 0.077 | 0.099 |
| | | Neural-B3 | 0.066 | 0.089 | 0.115 |
| summeval | relevance | Neural | 0.194 | 0.192 | 0.269 |
| | | Neural-B0 | 0.173 | 0.171 | 0.240 |
| | | Neural-B1 | 0.172 | 0.171 | 0.240 |
| | | Neural-B2 | 0.175 | 0.173 | 0.243 |
| | | Neural | 0.181 | 0.180 | 0.251 |

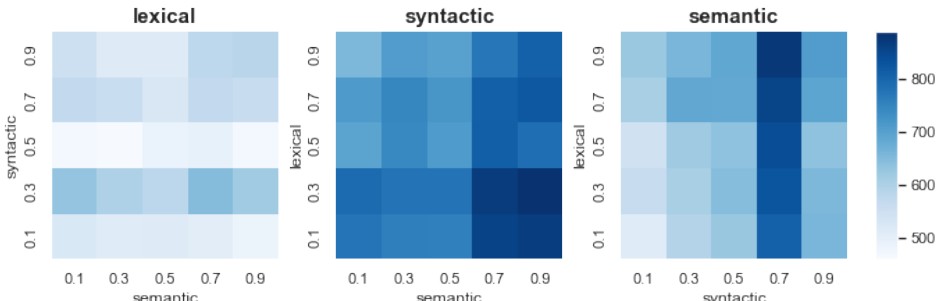

Figure 11: Number of control-generated phrases - for lexical similarity (left), syntactic similarity (middle), and semantic similarity (right). The X and Y axes show the similarity by the fixed qualities. The original phrases are from MRPC.

    parameters fixed). If increasing of the selected control parameter by 0.05 to the next value does not change the generated phrase, the generated phrase is ignored. The control parameters for the remaining two qualities remain fixed.

2. We obtained the embeddings of the original and of the generated phrases.
3. For each generated phrase, the embedding-based score was calculated as a dot-product between its embedding and the the embedding of the corresponding original phrase.
4. A correlation (Kendall Tau, variant c) was calculated between the embedding-based score and the generation control parameter.

Since we ignored duplicated generation phrases while increasing the control parameter along the values 0.05, 0.10, 0.15, ..., 0.95, there are around 300 - 900 scored phrases for each evaluation. We show the number of the generated phrases in Figures 11 and 12. At any choice of the fixed control

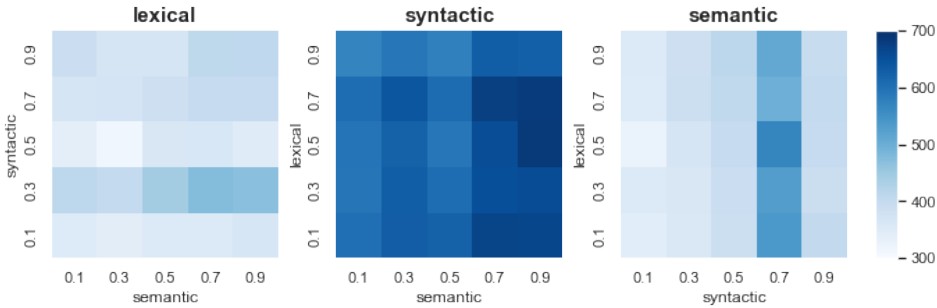

Figure 12: Number of control-generated phrases - for lexical similarity (left), syntactic similarity (middle), and semantic similarity (right). The X and Y axes show the similarity by the fixed qualities. The original phrases are from STS.

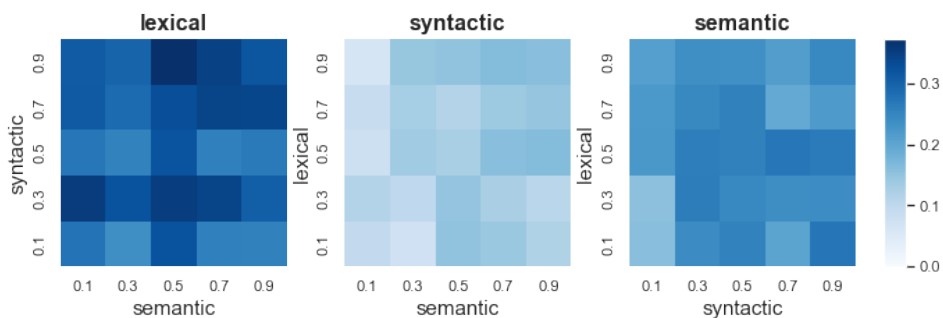

Figure 13: Evaluation of neural embeddings on short generated phrases with controlled similarity. The original phrases are from STS. Each cell represents Kendall Tau (variant c) correlation of the embedding-based score with a selected kind of similarity: lexical similarity on the left heatmap, syntactic similarity on the middle heatmap, and semantic similarity on the right heatmap. The X and Y axes show the similarity by the fixed qualities.

parameters, the number of generated phrases is not less than 300. All the correlations shown in Section 3.1 have p-value far below 0.05.

# E  CORRELATION OF EMBEDDING-BASED SCORES WITH SIMILARITY OF SHORT CONTROL-GENERATED SENTENCES.

In Section 3.1 we have considered correlations of embedding-based scores with similarity of sentences taken from MRPC dataset. MRPC sentences are typically long. Here we repeat the same review for shorter sentences. As our original sentences we select from STS test subset (Cer et al., 2017; May, 2021)[22] 100 first sentences of length between 20 and 40 characters, and generate the results in the same manner as was done in Section 3.1. The resulting correlations for neural embeddings are shown in Figure 13.

The corresponding ratios of the correlations of neural embedding-based score to the correlations based on other embeddings are shown in Figures 14, 15, 16, 17 and 18. Again, as in Section 3.1, neural embeddings have stronger correlations with semantic and lexical similarities, but now also, in comparison with some of the embeddings, stronger correlations with syntactic similarity.

---

[22]https://huggingface.co/datasets/stsb_multi_mt

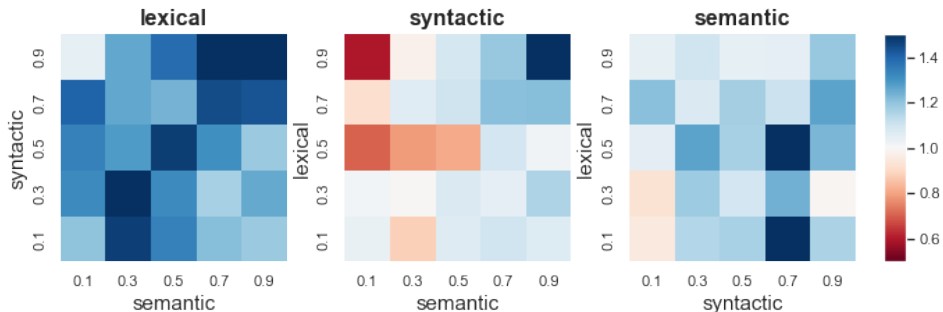

Figure 14: Ratio of correlations of neural embeddings to correlations of all-mpnet-base-v2 embeddings.

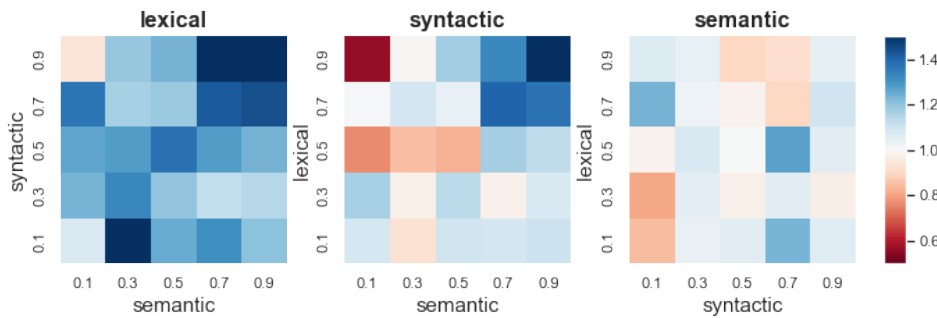

Figure 15: Ratio of correlations of neural embeddings to correlations of all-MiniLM-L6-v2 embeddings.

## F    UNNORMALIZED SENTENCE EMBEDDINGS

In our evaluations in Section 3 we used normalized embeddings (hence dot-product equal to cosine). Here we point out that having unnormalized embeddings for the models all-mpnet-base-v2, all-MiniLM-L6-v2 and LaBSE does not affect much our observations.

The count of errors (the column "wrong" in Table 1 changes when these embeddings are unnormalized by not more than 3 errors. The only difference for Table 3 is that unnormalized embeddings of model all-MiniLM-L6-v2 get the count "wrong" 736307 instead of 736308. And the correlations in Table 4 remain the same within the shown precision.

## G    NEURAL VS HIDDEN LAYER EMBEDDINGS

Neural embeddings, introduced in Section 2 differ from the commonly used 'hidden layer' embeddings in four factors or steps:

1. Micro-tuning: the model gets tuned on extremely small 'dataset', produced from a single sample.
2. Measuring the difference: The embedding is not simply taken from a (micro-tuned) model, but obtained as a difference between the embeddings from the micro-tuned model and the original model.
3. Using weights: The embedding is taken not from an output of a hidden layer, but from the weights of the model.
4. Combining layers: The results from several layers are concatenated.

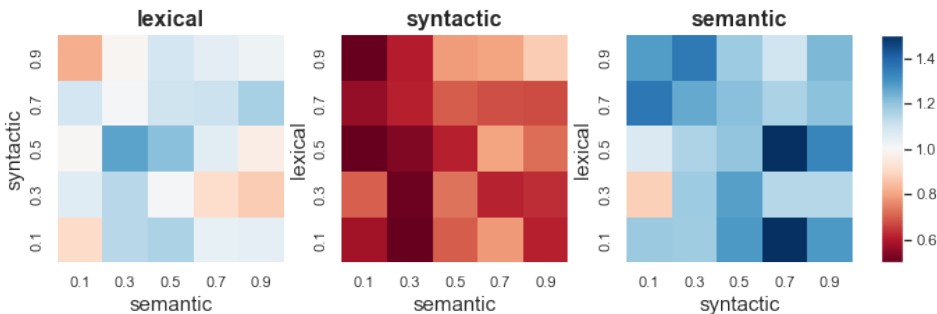

Figure 16: Ratio of correlations of neural embeddings to correlations of LaBSE embeddings.

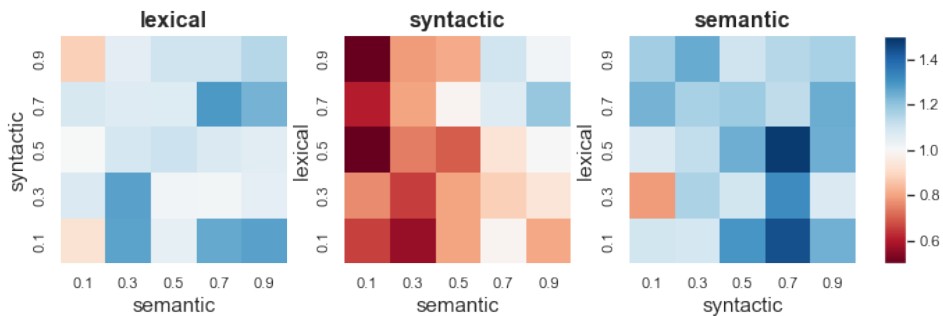

Figure 17: Ratio of correlations of neural embeddings to correlations of SGPT-125M embeddings.

Do all these three factors necessary? And if yes, how important is each of them? In order to answer these questions, we compare here the neural embeddings with their simplified versions, stripped from one or more of the factors. These are the simplified versions:

1. cls: embedding picked up from CLS token on output of the BERT last transformer block (block 11).
2. avg: average of embeddings picked up from all tokens of the text on output of the last block.
3. cls-tuned: The same as 'cls', but obtained from micro-tuned model.
4. avg-tuned: The same as 'avg', but obtained from micro-tuned model.
5. cls-diff: Obtained as the difference between 'cls-tuned' and 'cls'.
6. avg-diff: Obtained as the difference between 'avg-tuned' and 'avg'.
7. neural-1: Neural embedding using just one layer, specifically the layer (by huggigface transformers notations) 'bert.encoder.layer.11.output.dense.bias'.

For fair comparison, here the neural embeddings are also taken from block 11 (not from cls layers):

1. $L_0 = bert.encoder.layer.11.output.LayerNorm.weight$
2. $L_1 = bert.encoder.layer.11.output.LayerNorm.bias$
3. $L_2 = bert.encoder.layer.11.output.dense.bias$

Thus, in our simple collection of embeddings here, the 'cls' and 'avg' represent the usual embeddings. The 'cls-tuned' and 'avg-tuned' represent embeddings that included the first step toward the neural embeddings: micro-tuning. The 'cls-diff' and 'avg-diff' represent embeddings that included the second step toward neural embeddings: measuring the difference. Finally, the 'neural-1' represents the neural embedding with single layer. Notice that we selected the most valuable layer, accordingly to the ablation study in Appendix C.1.

The results of the evaluations are presented in Tables 13, 14, 15, 16. Micro-tuning was done with the same parameters as through the evaluations, Section 3, but here we used 5 (not 10) epochs. The tables show results for normalized versions of embeddings.

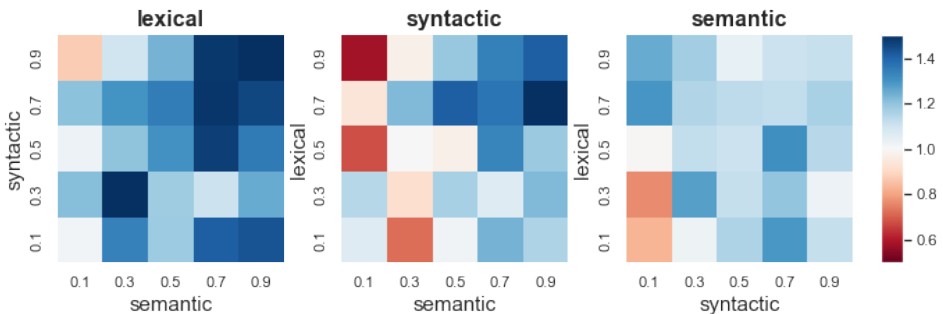

Figure 18: Ratio of correlations of neural embeddings to correlations of SGPT-5.8B embeddings.

For evaluations of Table 13, the unnormalized versions of embeddings make less errors only in the following cases, marked yellow: For dataset 'catasaurus', the embeddings 'cls-mtuned' make 57149 errors; for dataset 'asset/t', the embeddings 'cls' make 1672558 errors, and embeddings 'cls-mtuned' make 1232707 errors.

For evaluations of Tables 14 and 15, all unnormalized are worse (make more errors). In Table 16 we included unnormalized versions of embeddings (marked by suffix '-u'), in all cases when they do better than the normalized versions.

Overall, from the tables 13, 14, 15 and 16 we can conclude that all the factors matter, and micro-tuning is probably the strongest, most important factor.

## H    NEURAL AND SGPT EMBEDDINGS CONCATENATED

We observed in Section 3 that neural embeddings and SGPT behave differently: their errors do not overlap greatly, their average products have different ranges, and their correlations with text qualities are different. Here we suggest an additional illustration of this point. We create a simple 'ensemble' by concatenating neural embeddings and SGPT embeddings into a single embedding. An ensemble is expected to perform better than either of its components if the they perform not too differently but have different properties. Specifically we used here SGPT-125M embeddings.

In Table 17 we observe that this is indeed true. The exception is 'text-summ-g' where neural embedding was orders of magnitude better, 'cestwc/c' with almost a tie, and, strangely, catasaurus. In Table 18 for 'sts', SGPT was much better to start with. In Table 4 the neural and SGPT embeddings were almost equally good correlating with relevance, and in that case the concatenation is much stronger than either of them.

Interesting, even though evaluation of summary quality was not the purpose of the embeddings, neural embeddings correlations for consistency scores exceed correlations of all flavors of ROUGE (ROUGE-L, ROUGE-1, ROUGE-2 and ROUGE-3). Similarly, Neural+SGPT correlations are higher than all flavors of ROUGE for relevance, and SGPT correlations are higher for coherence.

Table 13: Neural embeddings vs micro-tuned embeddings vs usual embeddings. Triplets.

| dataset | embedding | error | total | wrong | same | diff |
|---|---|---|---|---|---|---|
| mrpc | neural | 6.7e-5 | 24472808 | 1639 | 0.54 | 0.03 |
| | neural-1 | 8.5e-5 | | 2076 | 0.57 | 0.04 |
| | cls | 1.4e-2 | | 333146 | 0.92 | 0.75 |
| | avg | 3.9e-4 | | 9637 | 0.92 | 0.62 |
| | cls-mtuned | 9.0e-3 | | 221189 | 0.92 | 0.73 |
| | avg-mtuned | 2.3e-4 | | 5608 | 0.90 | 0.56 |
| | cls-diff | 3.7e-4 | | 9044 | 0.56 | 0.06 |
| | avg-diff | 2.2e-4 | | 5432 | 0.58 | 0.05 |
| sts | neural | 1.0e-3 | 455624 | 471 | 0.53 | 0.04 |
| | neural-1 | 1.1e-3 | | 516 | 0.58 | 0.05 |
| | cls | 2.8e-2 | | 12805 | 0.92 | 0.73 |
| | avg | 3.8e-3 | | 1708 | 0.89 | 0.55 |
| | cls-mtuned | 2.2e-2 | | 9918 | 0.91 | 0.70 |
| | avg-mtuned | 2.8e-3 | | 1257 | 0.88 | 0.50 |
| | cls-diff | 2.1e-3 | | 975 | 0.56 | 0.05 |
| | avg-diff | 1.7e-3 | | 792 | 0.58 | 0.06 |
| catasaurus | neural | 7.1e-5 | 99980000 | 7082 | 0.78 | 0.02 |
| | neural-1 | 7.7e-5 | | 7716 | 0.81 | 0.02 |
| | cls | 1.1e-3 | | 107713 | 0.98 | 0.63 |
| | avg | 7.4e-5 | | 7369 | 0.98 | 0.51 |
| | cls-mtuned | 7.6e-4 | | 75988 | 0.97 | 0.60 |
| | avg-mtuned | 7.2e-5 | | 7146 | 0.97 | 0.46 |
| | cls-diff | 7.3e-5 | | 7313 | 0.80 | 0.03 |
| | avg-diff | 7.3e-5 | | 7333 | 0.82 | 0.03 |
| asset/t | neural | 4.9e-4 | 155511620 | 75916 | 0.50 | 0.02 |
| | neural-1 | 6.0e-4 | | 93684 | 0.54 | 0.02 |
| | cls | 1.1e-2 | | 1745865 | 0.92 | 0.69 |
| | avg | 3.9e-4 | | 60617 | 0.91 | 0.52 |
| | cls-mtuned | 8.6e-3 | | 1334770 | 0.91 | 0.67 |
| | avg-mtuned | 2.2e-4 | | 34829 | 0.89 | 0.47 |
| | cls-diff | 3.6e-3 | | 555409 | 0.52 | 0.03 |
| | avg-diff | 2.2e-3 | | 342306 | 0.54 | 0.02 |
| cestwc/t | neural | 4.1e-2 | 224550000 | 9243349 | 0.31 | 0.03 |
| | neural-1 | 4.5e-2 | | 10138993 | 0.34 | 0.04 |
| | cls | 6.3e-2 | | 14026873 | 0.94 | 0.87 |
| | avg | 2.8e-2 | | 6174293 | 0.78 | 0.53 |
| | cls-mtuned | 5.3e-2 | | 11884179 | 0.92 | 0.84 |
| | avg-mtuned | 2.4e-2 | | 5370379 | 0.75 | 0.48 |
| | cls-diff | 8.0e-2 | | 17848425 | 0.33 | 0.04 |
| | avg-diff | 5.3e-2 | | 11905894 | 0.34 | 0.03 |
| cestwc/q | neural | 1.1e-2 | 224550000 | 2389985 | 0.39 | 0.08 |
| | neural-1 | 1.2e-2 | | 2582021 | 0.42 | 0.09 |
| | cls | 7.1e-2 | | 15989315 | 0.94 | 0.89 |
| | avg | 1.1e-2 | | 2499481 | 0.85 | 0.64 |
| | cls-mtuned | 5.2e-2 | | 11637580 | 0.93 | 0.86 |
| | avg-mtuned | 8.3e-3 | | 1860801 | 0.83 | 0.60 |
| | cls-diff | 3.5e-2 | | 7795495 | 0.41 | 0.10 |
| | avg-diff | 2.1e-2 | | 4623758 | 0.44 | 0.11 |
| cestwc/c | neural | 1.1e-1 | 224550000 | 25060134 | 0.24 | 0.10 |
| | neural-1 | 1.2e-1 | 224550000 | 26195509 | 0.29 | 0.12 |
| | cls | 3.1e-1 | | 68860399 | 0.83 | 0.79 |
| | avg | 1.2e-1 | | 25818812 | 0.79 | 0.67 |
| | cls-mtuned | 2.8e-1 | | 63727774 | 0.82 | 0.77 |
| | avg-mtuned | 1.0e-1 | | 22631189 | 0.76 | 0.63 |
| | cls-diff | 1.6e-1 | | 35766534 | 0.26 | 0.11 |
| | avg-diff | 1.4e-1 | | 32400635 | 0.30 | 0.14 |

Table 14: Neural embeddings vs micro-tuned embeddings vs usual embeddings. Triplets. Summaries and texts.

| dataset | embedding | error | total | wrong | same | diff |
|---|---|---|---|---|---|---|
| text-summ-g | neural | 3.8e-5 | 2861100 | 109 | 0.42 | 0.03 |
| | neural-1 | 4.1e-4 | | 1162 | 0.42 | 0.03 |
| | cls | 5.6e-2 | | 160086 | 0.81 | 0.69 |
| | avg | 1.6e-2 | | 46442 | 0.91 | 0.76 |
| | cls-mtuned | 3.7e-2 | | 106778 | 0.80 | 0.65 |
| | avg-mtuned | 4.7e-3 | | 13322 | 0.87 | 0.67 |
| | cls-diff | 2.1e-3 | | 6051 | 0.42 | 0.04 |
| | avg-diff | 6.5e-4 | | 1854 | 0.44 | 0.03 |
| text-summ-r | neural | 6.9e-3 | 1197900 | 8277 | 0.26 | 0.03 |
| | neural-1 | 1.1e-2 | | 13520 | 0.27 | 0.02 |
| | cls | 9.5e-2 | | 113506 | 0.78 | 0.68 |
| | avg | 5.4e-2 | | 64511 | 0.86 | 0.73 |
| | cls-mtuned | 7.2e-2 | | 85916 | 0.76 | 0.64 |
| | avg-mtuned | 2.7e-2 | | 32740 | 0.81 | 0.64 |
| | cls-diff | 2.9e-2 | | 34862 | 0.27 | 0.04 |
| | avg-diff | 1.9e-2 | | 22154 | 0.28 | 0.02 |
| summ-g | neural | 5.6e-3 | 45777600 | 255665 | 0.44 | 0.03 |
| | neural-1 | 8.4e-3 | | 386119 | 0.46 | 0.04 |
| | cls | 4.6e-2 | | 2094610 | 0.86 | 0.71 |
| | avg | 8.2e-3 | | 373988 | 0.92 | 0.71 |
| | cls-mtuned | 3.6e-2 | | 1646157 | 0.85 | 0.68 |
| | avg-mtuned | 5.4e-3 | | 244771 | 0.89 | 0.64 |
| | cls-diff | 1.6e-2 | | 710337 | 0.46 | 0.05 |
| | avg-diff | 1.2e-2 | | 541905 | 0.47 | 0.04 |
| summ-r | neural | 3.6e-2 | 11979000 | 435033 | 0.22 | 0.03 |
| | neural-1 | 4.6e-2 | | 548970 | 0.25 | 0.04 |
| | cls | 9.5e-2 | | 1137445 | 0.83 | 0.73 |
| | avg | 3.6e-2 | | 432290 | 0.85 | 0.69 |
| | cls-mtuned | 7.9e-2 | | 942028 | 0.81 | 0.70 |
| | avg-mtuned | 2.7e-2 | | 327481 | 0.80 | 0.62 |
| | cls-diff | 7.9e-2 | | 946548 | 0.24 | 0.05 |
| | avg-diff | 7.2e-2 | | 862239 | 0.24 | 0.03 |

Table 15: Neural embeddings vs micro-tuned embeddings vs usual embeddings. Pairs similar and not similar.

| dataset | embedding | error | total | wrong | same | diff |
|---|---|---|---|---|---|---|
| mrpc | neural | 2.7e-1 | 2953956 | 785630 | 0.54 | 0.42 |
| | neural-1 | 2.7e-1 | | 784603 | 0.57 | 0.46 |
| | cls | 3.7e-1 | | 1100243 | 0.92 | 0.90 |
| | avg | 2.8e-1 | | 837001 | 0.92 | 0.88 |
| | cls-mtuned | 3.6e-1 | | 1056685 | 0.92 | 0.89 |
| | avg-mtuned | 2.7e-1 | | 788863 | 0.90 | 0.86 |
| | cls-diff | 2.8e-1 | | 830654 | 0.56 | 0.45 |
| | avg-diff | 2.7e-1 | | 796859 | 0.58 | 0.46 |
| sts | neural | 1.1e-1 | 180492 | 19194 | 0.53 | 0.30 |
| | neural-1 | 1.1e-1 | | 19686 | 0.58 | 0.34 |
| | cls | 3.5e-1 | | 62899 | 0.92 | 0.89 |
| | avg | 1.9e-1 | | 33342 | 0.89 | 0.79 |
| | cls-mtuned | 3.2e-1 | | 56930 | 0.91 | 0.88 |
| | avg-mtuned | 1.6e-1 | | 28524 | 0.88 | 0.77 |
| | cls-diff | 1.2e-1 | | 21103 | 0.56 | 0.33 |
| | avg-diff | 1.2e-1 | | 21167 | 0.58 | 0.34 |

Table 16: Neural embeddings vs micro-tuned embeddings vs usual embeddings.

| dataset | score | embedding | kendall-c | kendall-b | spearman |
|---|---|---|---|---|---|
| sts | similarity | neural | 0.468 | 0.471 | 0.646 |
| | | neural-1 | 0.469 | 0.472 | 0.645 |
| | | cls | 0.137 | 0.138 | 0.203 |
| | | avg | 0.346 | 0.348 | 0.497 |
| | | cls-mtuned | 0.173 | 0.174 | 0.255 |
| | | avg-mtuned | 0.383 | 0.385 | 0.544 |
| | | cls-diff | 0.443 | 0.446 | 0.618 |
| | | avg-diff | 0.456 | 0.459 | 0.630 |
| summeval | coherence | neural | 0.105 | 0.102 | 0.143 |
| | | neural-1 | 0.099 | 0.096 | 0.136 |
| | | cls | 0.007 | 0.007 | 0.010 |
| | | avg | 0.088 | 0.085 | 0.119 |
| | | cls-mtuned | 0.016 | 0.016 | 0.022 |
| | | avg-mtuned | 0.086 | 0.083 | 0.118 |
| | | cls-diff | 0.069 | 0.067 | 0.095 |
| | | avg-diff | 0.082 | 0.080 | 0.113 |
| summeval | consistency | neural | 0.096 | 0.153 | 0.194 |
| | | neural-1 | 0.075 | 0.119 | 0.151 |
| | | cls | 0.036 | 0.058 | 0.073 |
| | | cls-u | 0.047 | 0.074 | 0.095 |
| | | avg | 0.079 | 0.125 | 0.159 |
| | | cls-mtuned | 0.045 | 0.072 | 0.091 |
| | | cls-mtuned-u | 0.052 | 0.083 | 0.106 |
| | | avg-mtuned | 0.085 | 0.135 | 0.171 |
| | | cls-diff | 0.069 | 0.109 | 0.140 |
| | | avg-diff | 0.069 | 0.111 | 0.141 |
| summeval | fluency | neural | 0.058 | 0.078 | 0.100 |
| | | neural-1 | 0.044 | 0.059 | 0.076 |
| | | cls | 0.048 | 0.064 | 0.083 |
| | | cls-u | 0.073 | 0.098 | 0.127 |
| | | avg | 0.047 | 0.063 | 0.081 |
| | | avg-u | 0.063 | 0.084 | 0.109 |
| | | cls-mtuned | 0.053 | 0.071 | 0.092 |
| | | cls-mtuned-u | 0.076 | 0.101 | 0.131 |
| | | avg-mtuned | 0.056 | 0.075 | 0.096 |
| | | avg-mtuned-u | 0.070 | 0.094 | 0.121 |
| | | cls-diff | 0.034 | 0.045 | 0.058 |
| | | avg-diff | 0.042 | 0.056 | 0.073 |
| summeval | relevance | neural | 0.198 | 0.196 | 0.274 |
| | | neural-1 | 0.184 | 0.182 | 0.255 |
| | | cls | 0.077 | 0.077 | 0.109 |
| | | avg | 0.186 | 0.184 | 0.258 |
| | | cls-mtuned | 0.088 | 0.087 | 0.124 |
| | | avg-mtuned | 0.185 | 0.183 | 0.258 |
| | | cls-diff | 0.179 | 0.177 | 0.247 |
| | | avg-diff | 0.181 | 0.180 | 0.251 |

Table 17: Exampe of concatenation, measured against the triplets (Eq.2). Neural, SGPT-125M, and the concatenated embedding 'Neural+SGPT' of Neural and SGPT-125M embeddings. A row with lowest error is marked cyan.

| dataset | embedding | error | total | wrong | $I$ | same | diff |
|---|---|---|---|---|---|---|---|
| mrpc | SGPT | 7.9e-5 | 24472808 | 1945 | | 0.86 | 0.18 |
| | Neural | 6.1e-5 | | 1491 | 0.78 | 0.54 | 0.22 |
| | Neural+SGPT | 5.7e-5 | | 1394 | 0.90 | 0.70 | 0.10 |
| sts | SGPT | 1.0e-3 | 455624 | 460 | | 0.85 | 0.09 |
| | Neural | 1.0e-3 | | 455 | 0.69 | 0.53 | 0.03 |
| | Neural+SGPT | 9.0e-4 | | 412 | 0.86 | 0.69 | 0.06 |
| catasaurus | SGPT | 6.3e-5 | 99980000 | 6308 | | 0.97 | 0.19 |
| | Neural | 7.0e-5 | | 6998 | 0.88 | 0.79 | 0.02 |
| | Neural+SGPT | 6.8e-5 | | 6751 | 0.91 | 0.88 | 0.10 |
| asset/t | SGPT | 9.7e-5 | 155511620 | 15028 | | 0.86 | 0.13 |
| | Neural | 4.8e-4 | | 74731 | 0.31 | 0.51 | 0.01 |
| cestwc/t | SGPT | 1.9e-2 | 224550000 | 4282171 | | 0.68 | 0.15 |
| | Neural | 4.2e-2 | | 9509234 | 0.42 | 0.31 | 0.03 |
| | Neural+SGPT | 1.7e-2 | | 3775736 | 0.80 | 0.50 | 0.09 |
| cestwc/q | SGPT | 2.1e-3 | 224550000 | 479294 | | 0.77 | 0.22 |
| | Neural | 1.0e-2 | | 2280159 | 0.43 | 0.38 | 0.07 |
| | Neural+SGPT | 2.0e-3 | | 439138 | 0.66 | 0.57 | 0.15 |
| cestwc/c | SGPT | 4.4e-2 | 224550000 | 9920527 | | 0.60 | 0.17 |
| | Neural | 1.1e-1 | | 24563942 | 0.57 | 0.23 | 0.08 |
| | Neural+SGPT | 4.4e-2 | | 9985684 | 0.82 | 0.42 | 0.13 |
| text-summ-g | SGPT | 8.0e-3 | 2861100 | 22771 | | 0.70 | 0.22 |
| | Neural | 1.5e-5 | | 44 | 0.45 | 0.42 | 0.03 |
| | Neural+SGPT | 3.7e-4 | | 1068 | 1.00 | 0.56 | 0.12 |
| text-summ-r | SGPT | 1.1e-2 | 1197900 | 12967 | | 0.71 | 0.21 |
| | Neural | 6.7e-3 | | 7988 | 0.16 | 0.27 | 0.03 |
| | Neural+SGPT | 4.4e-3 | | 5238 | 0.94 | 0.49 | 0.12 |
| summ-g | SGPT | 6.3e-3 | 45777600 | 287978 | | 0.78 | 0.25 |
| | Neural | 4.4e-3 | | 203275 | 0.20 | 0.45 | 0.03 |
| | Neural+SGPT | 2.6e-3 | | 118566 | 0.93 | 0.61 | 0.14 |
| summ-r | SGPT | 1.6e-2 | 11979000 | 190809 | | 0.66 | 0.18 |
| | Neural | 3.7e-2 | | 440827 | 0.34 | 0.22 | 0.02 |
| | Neural+SGPT | 1.2e-2 | | 143632 | 0.86 | 0.44 | 0.10 |

Table 18: Exampe of concatenation of Neural and SGPT-125M. Measured on pairs (similar vs not similar).

| dataset | embedding | error | total | wrong | $I$ | same | diff |
|---|---|---|---|---|---|---|---|
| mrpc | SGPT | 2.4e-1 | 2953956 | 695606 | | 0.86 | 0.76 |
| | Neural | 2.6e-1 | | 772375 | 0.62 | 0.54 | 0.42 |
| | Neural+SGPT | 2.3e-1 | | 685446 | 0.77 | 0.70 | 0.59 |
| sts | SGPT | 4.1e-2 | 180492 | 7332 | | 0.85 | 0.48 |
| | Neural | 9.9e-2 | | 17893 | 0.56 | 0.53 | 0.29 |
| | Neural+SGPT | 4.8e-2 | | 8735 | 0.70 | 0.69 | 0.38 |

Table 19: Exampe of concatenation of Neural and SGPT-125M. Correlations with human scores.

| dataset | score | embedding | kendall-c | kendall-b | spearman |
|---------|-------|-----------|-----------|-----------|----------|
| sts | similarity | SGPT | 0.608 | 0.612 | 0.795 |
| | | Neural | 0.478 | 0.482 | 0.658 |
| | | Neural+SGPT | 0.592 | 0.596 | 0.779 |
| summeval | coherence | SGPT | 0.156 | 0.151 | 0.216 |
| | | Neural | 0.100 | 0.097 | 0.137 |
| | | Neural+SGPT | 0.156 | 0.151 | 0.215 |
| summeval | consistency | SGPT | 0.011 | 0.017 | 0.022 |
| | | Neural | 0.098 | 0.157 | 0.200 |
| | | Neural+SGPT | 0.068 | 0.108 | 0.138 |
| summeval | fluency | SGPT | -0.004 | -0.006 | -0.007 |
| | | Neural | 0.063 | 0.084 | 0.109 |
| | | Neural+SGPT | 0.038 | 0.051 | 0.065 |
| summeval | relevance | SGPT | 0.198 | 0.196 | 0.275 |
| | | Neural | 0.190 | 0.188 | 0.263 |
| | | Neural+SGPT | 0.233 | 0.231 | 0.324 |

