# OpenReview forum: "Neural Embeddings for Text"
_ICLR.cc/2023/Conference — Submitted to ICLR 2023_

### Official Review · Reviewer_JM7i · 2022-10-17

**Confidence:** 5
**Correctness:** 2
**Technical Novelty And Significance:** 2
**Empirical Novelty And Significance:** 2
**Recommendation:** 3

**Clarity, Quality, Novelty And Reproducibility:**

The clarity of manuscript is not low, and is easy to understand.
While the reproducibility is also not low,
both its quality and novelty are not high.
That leads to the following review decision.

**Strength And Weaknesses:**

Strength\
*Authors take an empirical and theoretical approach\
*Compare the smaller models with the larger model, SGPT\
*Human analysis

Weaknesses\
*The theoretical background of the proposed approach is weak.\
　The theoretical background and reasons why tuning and masking strategies can acquire semantic representation of texts are unclear. \
*Authors' improvement is unclear in applying the micro-tuning\
　What is the new rationale for the application of a periodic masking strategy in BERT? \
　Why you use BERT instead of GPT? \
*Insufficient comparison of technical aspects \
　Authors apply only the product of the embeddings to compare the semantic similarity  in Eq (2). \
　Is the product of embeddings more common than the difference in distance in triplets? \
*No ablation analysis \
　Is it fair to compare GPT-base (decoder) and Bert-base (encoder) models in the applied task? \
    If we don't apply  a periodic masking strategy to GPT-base model,  \
    isn't BERT-base models, not the masking strategy, affecting the outcome?

**Summary Of The Paper:**

This paper proposes a text embedding that deeply represents semantic meaning,
and calls it a neural embedding.
While the conventional text embeddings use the vector output of a pre-trained language model,
the neural embedding learns the text and literally pick its brain, and takes the actual weights of the model’s neurons to generate a vector.
More precisely, it allows to learn model’s neurons from text, and may reflect deeper semantic features of the text.
Authors compare GPT sentence (SGPT) embedding over  several datasets and show that 1) the neural embeddings are comparable to SGPT embeddings, and 2) its errors and qualities are often different.


**Summary Of The Review:**

Although author's motivation is interesting,
they do not provide its theoretical basis and show that comparisons are insufficient.
This is the reason why I can not appreciate it.

---

### Official Review · Reviewer_gPmr · 2022-10-24

**Confidence:** 4
**Correctness:** 2
**Technical Novelty And Significance:** 4
**Empirical Novelty And Significance:** 4
**Recommendation:** 3

**Clarity, Quality, Novelty And Reproducibility:**

The paper presents a novel idea, and no obvious mistakes were made in the evaluation. It would be fairly easy to repeat the experiments with the provided code and exhaustive descriptions in the paper.

The writing could use improvement in many places:
* The description of L_1 ... L_3 are based on their naming conventions in the Huggingface Transformer library. First, this is not understandable to people who are not familiar with this library. A description of what these weights refer to is needed. Second, the choice of layers seems arbitrary. Why use the bias vector of the linear projection layer but not the weight matrix? Why are the weights of LayerNorm included? A justification and explanation of this selection would be helpful.
* Table captions need to go above the table according to the ICLR style guidelines.
* At the end of Section 2, there is a reference to a lower quality neural embedding model (ablation on the training objective) supposedly evaluated in Section 3. However, in Section 3, no such evaluation is to be found.
* Table 1: The columns total and wrong seem to add no value, but only contain large, uninformative writers. For all practical purposes, isn't the 'error' column more than enough?
* You say that for 'text-summ-g' neural embeddings are sensitive to the random seed. Why do you only report the sensitivity here? Shouldn't it be senstitive to the random seed on all datasets?
* Figures are no vector graphics, bad quality when zooming in
* Section 3.4: The quality-controlled paraphrase generation model is not described in enough detail. The description is not exhaustive enough to fully understand what Section 3.4. is measuring exactly.

**Strength And Weaknesses:**

Strengths:
* sensible and interesting idea, easy to understand
* comparison on a lot of text similarity benchmarks
* promising results, as the proposed method seems complementary to the baseline in many regards

Weaknesses:
* no evaluation of the time complexity in comparison to the baseline. Only the per example time of the proposed method is evaluated (in appendix), but a reference point is missing. This makes it difficult to judge the practicality of the approach, and (probably) hides its high computational cost.
* no controlled experiment / ablation: BERT is used for creating the neural embeddings, so it would be logical to use averaged BERT embeddings as a baseline, which would constitute a controlled experiment. While comparison to SOTA is nice (as an upper bound), you cannot rule out that the observed qualitative differences between embedding methods are actually due to the underlying model architecture or training data, rather than the way embeddings are created. This weakens the support for the main claim.
* writing could be improved. There are many places that appear a bit sloppy (but could be fixed easily); details in the Clarity section below.
* the method includes a masking scheme that deviates from the original BERT masking objective. As no ablations or explanations are provided, it remains unclear how this specific training objective influences the performance of the model. Could it be responsible for the observed qualitative differences?


**Summary Of The Paper:**

The paper proposes a new way of representing a text as a single fixed-size vector. While most text embedding methods use model activations to represent the input text, the proposed method computes the difference in model parameters that is induced by finetuning on a single text input towards. The finetuning task is a custom masked language modeling objective.
The new method is compared to a recent SOTA model that uses activations as embeddings. Through exhaustive text similarity evaluations, the new model is shown to be qualitatively different from the baseline, exceeding it in performance in several regards.

**Summary Of The Review:**

The paper presents a worthwhile and novel idea that shows promising initial results against a state-of-the-art baseline. However, only a controlled experiment can support the main claim that neural embeddings learn something qualitatively different from conventional embeddings. Moreover, the reader learns only little about what makes the model perform due to a lack of ablations on e.g. the training objective. I therefore recommend rejection.

---

### Official Review · Reviewer_SjTe · 2022-10-25

**Confidence:** 5
**Correctness:** 3
**Technical Novelty And Significance:** 1
**Empirical Novelty And Significance:** 1
**Recommendation:** 3

**Clarity, Quality, Novelty And Reproducibility:**

Clarify: The idea presented in this work is clear to understand.
Quality: Low. See above.
Novelty: The idea appears to be new, but the work does not justify why the proposed method is desired among alternatives.
Reproducibility: Code will be released according to the authors.

**Strength And Weaknesses:**

Strength:
It is an interesting idea to represent text by the actual neural network weights (or the change of it after tuning in specific).

Weakness:
The paper fails to establish why this is a desired method, for the following reasons.

1. The benchmarks (ranking triplets) used are quite elementary. Tasks close to real world scenarios should be used (e.g. retrieval on MS MARCO and other tasks in SGPT and related works).

2. It shows that the proposed embeddings method (110M parameters) performs comparably to SGPT (5.8B parameters), but this does not answer the question whether a smaller SGPT model can achieve similar performance, and whether a larger model with the proposed method can outperform SGPT.

3. Conventional embeddings are defined as the output of a trained neural network. So the embeddings can be efficiently generated when running inference. However the proposed embedding requires running training. This can be far more expensive and hard to compute even if the network is much smaller in size.

**Summary Of The Paper:**

This work proposes a new type of embeddings of text. Instead of relying on the output of the neural network, the proposed embedding is generated by the actual weights of the network, when it is tuned with the specific content/text. The work shows that the embeddings generated this way capture semantic differences between text -- texts with similar meanings are closer to each other in the embedding space.



**Summary Of The Review:**

It is an interesting idea to consider the change of neural network weights in the training process as embeddings. Understanding of the semantics represented by the neural network weights is also a fundamental problem of interests to the community.  However the work fails to establish why the proposed method is superior in comparison with conventional embedding methods.

---

### Official Review · Reviewer_fZyy · 2022-10-25

**Confidence:** 4
**Correctness:** 3
**Technical Novelty And Significance:** 2
**Empirical Novelty And Significance:** 2
**Recommendation:** 5

**Clarity, Quality, Novelty And Reproducibility:**

Quality:
It is a bit hard for me to find actual usefulness or effectiveness of the proposed method.
However, it may be possible that I miss something.

Clarity:
It seems that there are several notations that are not clearly explained in this paper.
For example, || in Equation 1.
Moreover, I guess W represents the matrix. In this case, what do the absolute value of the matrix represent, such as |W| represents, and what is the meaning of E/|E| ?

Originality:
The originality seems very high since, to the best of my knowledge, I have never seen a similar method before.



**Details Of Ethics Concerns:**

I found no ethical concerns.

**Strength And Weaknesses:**

Strength:
* The proposed method seems novel.
* The task this paper tackles is an essential technology in the NLP field.


Weaknesses:
* I agree with the importance of developing a better text representation. However, the paper does not describe the merits of using the proposed method compared with other existing methods, such as SGPT. Therefore, I am wondering what the advantages of the proposed method are.
* If I do not miss something, there is no explicit discussion of theoretical justification for the proposed method. Therefore, I am not convinced of the effectiveness of the proposed method.


**Summary Of The Paper:**

This paper focuses on developing a new type of embeddings for texts (sentences or documents, but not words).
The proposed method's basic idea is to consider actual weights in the trained models (seem to assume language models).
Then, the method computes the weight difference between the original and micro-tuned ones (fine-tuning one sample only).
The method is evaluated on several benchmark datasets for semantic computing similarity among texts.
The experimental results show comparable results with embeddings obtained from GPT-3.







**Summary Of The Review:**

Overall, while the originality of the proposed method is high, the advantage of the proposed method seems to be limited.
This is because, as shown in the experiments and ponted out by the authors, the performance of the proposed method does not clearly surpass that of SGPT.
The authors claim that the proposed method can be compared using a smaller model but seems to require additional computational costs for micro-tuning.
This property seems to limit the usefulness of the proposed method.
Therefore, I lean toward not recommending this paper to be accepted to the conference.

---

### Author Response · Authors · 2022-11-18
**Updated version submitted**

The revised and updated version is uploaded, thanks to all reviewers for helpful comments. The paper (in the first 9 pages or in Appendixes) accounted for the most of the comments, except the following:

1. We would like to keep the table columns ‘total’ and ‘wrong’ because they provide exact data about the number of triplets and number of errors made. On the other hand, the ‘error’ column allows quickly grasping the accuracy, while not taking much space.

2. We left evaluation by distance (instead of by the product) for the future. (Of course this makes a difference only for unnormalized embeddings.)

---

### Author Response · Authors · 2023-02-03
**In https://arxiv.org/abs/2208.08386**

The paper is also in https://arxiv.org/abs/2208.08386

If curious about the factors influencing the quality of neural embeddings, please see Appendices C.1 (ablation studies with respect to layers), C.2 (ablation studies with respect to mask-blueprints) and G (ablation studies with respect to the factors of the approach).

---

### Decision · Program_Chairs · 2023-01-20

**Decision:**

Reject

**Justification For Why Not Higher Score:**

None of the reviewers would like to recommend this paper to be accepted. The authors did not respond to the reviewers' comments during the rebuttal.

**Justification For Why Not Lower Score:**

N/A

**Metareview: Summary, Strengths And Weaknesses:**

This paper proposes a new method for learning representations for texts. The idea was to use the weights from trained neural language models as representations. The authors compared their representations with those produced by existing models, arguing that the proposed method is superior to those previous approaches.

The reviewers largely agree that the idea seems interesting, and the results show that the proposed approach works well for the task to some extent.

There are several concerns over the current work. One major complaint is that the work needs to offer sufficient insights into why the community shall adopt their proposed approach. Specifically, despite the large number of experimental results reported in the paper, the evaluations are insufficient. The authors should consider adding some controlled experiments and ablation studies to examine where the approach's effectiveness comes from carefully. It would be good to have theoretical discussions to understand the benefits of the proposed method.